# COUNTERFACTUAL DENSITY ESTIMATION USING KERNEL STEIN DISCREPANCIES

**Diego Martinez-Taboada**
Department of Statistics & Data Science
Carnegie Mellon University
Pittsburgh, PA 15213, USA
diegomar@andrew.cmu.edu

**Edward H. Kennedy**
Department of Statistics & Data Science
Carnegie Mellon University
Pittsburgh, PA 15213, USA
edward@stat.cmu.edu

## ABSTRACT

Causal effects are usually studied in terms of the means of counterfactual distributions, which may be insufficient in many scenarios. Given a class of densities known up to normalizing constants, we propose to model counterfactual distributions by minimizing kernel Stein discrepancies in a doubly robust manner. This enables the estimation of counterfactuals over large classes of distributions while exploiting the desired double robustness. We present a theoretical analysis of the proposed estimator, providing sufficient conditions for consistency and asymptotic normality, as well as an examination of its empirical performance.

## 1 INTRODUCTION

Causal targets examine the outcomes that might have occurred if a specific treatment had been administered to a group of individuals. Generally, only the expected value of these outcomes is analyzed, as seen in the widely used average treatment effect. Nonetheless, focusing solely on means proves insufficient in many scenarios. Important attributes of the counterfactuals, such as their variance or skewness, are often disregarded. Modelling the entire distribution gives a complete picture of the counterfactual mechanisms, which opens the door to a richer analysis. For example, a multimodal structure in the counterfactual may indicate the presence of distinct subgroups with varying responses to the treatment (Kennedy et al., 2021). This profound understanding of the counterfactual may be ultimately exploited in the design of new treatments.

In order to model a distribution, one may consider estimating either its cumulative distribution function (CDF) or its probability density function (PDF). While the statistical analysis of the former is generally more straightforward, the latter tends to be more attractive for practitioners given its appealing interpretability. Although density estimation may be conducted non-parametrically, modelling probability density functions based on a parametric class of distributions has garnered significant attention. These models allow easy integration of prior data knowledge and offer interpretable parameters describing distribution characteristics.

However, a number of interesting parametric density functions are only known up to their normalizing constant. Energy-based models (LeCun et al., 2006) establish probability density functions $q_\theta(y) \propto \exp(-E_\theta(y))$, where $E_\theta$ is a parametrized *potential* fulfilling $\int \exp(-E_\theta(y))dy < \infty$. Sometimes, the energy is expressed as a combination of hidden and visible variables, namely product of experts or restricted Boltzmann machines (Ackley et al., 1985; Hinton, 2002; 2010). In contrast, energy-based models may link inputs directly to outputs (Mnih & Hinton, 2005; Hinton et al., 2006). Gibbs distributions and Markov random fields (Geman & Geman, 1984; Clifford, 1990), as well as exponential random graph models (Robins et al., 2007; Lusher et al., 2013) are also examples of this form of parameterization. We highlight the generality of energy-based models, which allow for modelling distributions with outstanding flexibility.

Generally, the primary difficulty in manipulating energy-based models stems from the need to precisely estimate the normalizing constant. Nonetheless, this challenge may be circumvented by using the so-called kernel Stein discrepancies, which only require the computation of the score function $\nabla_x \log q_\theta(x)$. Given a class of distributions $\mathcal{Q} = \{q_\theta\}_{\theta \in \Theta}$ and samples from a base distribution $Q$, one may model the latter by the distribution $q_{\theta_n}$ that minimizes its kernel Stein discrepancy with

respect to the empirical distribution $Q_n$. However, these minimum kernel Stein discrepancy estimators have not been explored in the challenging counterfactual setting, where outcomes are not always observed, and the treatment assignment procedure ought to be taken into account.

In this work, we propose to model counterfactuals by minimizing kernel Stein discrepancies in a doubly robust manner. While the presented estimator retains the desired properties of double robustness, it enables flexible modelling of the counterfactual via density functions with normalizing constants that need not be specified. Our contributions are two-fold. First, we present a novel estimator for modelling counterfactual distributions given a parametric class of distributions, along with its theoretical analysis. We provide sufficient conditions for both consistency and asymptotic normality. Second, we illustrate the empirical performance of the estimator in a variety of scenarios.

## 2 RELATED WORK

Counterfactual distribution estimation has been mainly addressed based on CDF approximation (Abadie, 2002; Chernozhukov & Hansen, 2005; Chernozhukov et al., 2013; Díaz, 2017), where the predominant approach reduces to counterfactual mean estimation. In contrast, counterfactual PDF estimation generally relies on kernel smoothing (Robins & Rotnitzky, 2001; Kim et al., 2018) or projecting the empirical distribution onto a finite-dimensional model using $f$-divergences or $L^p$ norms (Westling & Carone, 2020; Kennedy et al., 2021; Melnychuk et al., 2023). We highlight that none of these alternatives can in principle handle families of densities with unknown normalizing constants. For instance, using the general $f$-divergences or $L^p$ norms in the projection stage of the estimation (Kennedy et al., 2021) requires access to the exact evaluation of the densities, which includes the normalizing constants. This is precisely what motivated the authors of this contribution to explore kernel Stein discrepancies in the counterfactual setting.

Kernel Stein discrepancies (KSD), which build on the general Stein's method (Stein, 1972; Gorham & Mackey, 2015), were first introduced for conducting goodness-of-fit tests and sample quality analysis (Liu et al., 2016; Chwialkowski et al., 2016; Gorham & Mackey, 2017); they may be understood as a kernelized version of score-matching divergence (Hyvärinen & Dayan, 2005). Minimum kernel Stein discrepancy (MKSD) estimators were subsequently proposed (Barp et al., 2019; Matsubara et al., 2022), which project the empirical distribution onto a finite-dimensional model using the KSD. We highlight that MKSD estimators had not been proposed in the counterfactual settings prior to this work. Lam & Zhang (2023) suggested a doubly-robust procedure to estimate expectations via Monte Carlo simulation using KSD, but their motivation and analysis significantly diverge from our own contributions: while MKSD estimators minimise the KSD over a class of distributions, (quasi) Monte Carlo methods exploit KSD by transporting the sampling distribution (Oates et al., 2017; Fisher et al., 2021; Korba et al., 2021). We refer the reader to Anastasiou et al. (2023) for a review on Stein's methods, and to Oates et al. (2022) for an overview of MKSD estimators.

Kernel methods have been gaining interest in causal inference for assessing whole counterfactual distributions. In order to test for (conditional) distributional treatment effects, Muandet et al. (2021) and Park et al. (2021) made use of kernel mean embeddings via inverse propensity weighting and plug-in estimators, respectively. This line of work was later generalized to doubly robust estimators by Fawkes et al. (2022) and Martinez-Taboada et al. (2023). Beyond distributional representation, kernel regressors have have found extensive use in counterfactual tasks (Singh et al., 2019; 2020; 2021; Zhu et al., 2022).

## 3 BACKGROUND

Let $(X, A, Y) \sim P$, where $X \in \mathcal{X}$, $Y \in \mathcal{Y} \subset \mathbb{R}^d$, and $A \in \{0, 1\}$ represent the covariates, outcome, and binary treatment respectively. We frame the problem in terms of the potential outcome framework (Rubin, 2005; Imbens & Rubin, 2015), assuming A1) $Y = AY^1 + (1 - A)Y^0$ (where $Y^1 \sim Q^1$ and $Y^0 \sim Q^0$ are the *potential outcomes* or *counterfactuals*), A2) $Y^0, Y^1 \perp\!\!\!\perp A \mid X$, A3) $\epsilon < \pi(X) := P_{A|X}(A = 1|X) < 1 - \epsilon$ almost surely for some $\epsilon > 0$.

Conditions A1-A3 are ubiquitous in causal inference, but other identifiability assumptions are also possible. Condition A1 holds true when potential outcomes are exclusively determined by an indi-

vidual's own treatment (i.e., no interference), and Condition A2 applies when there are no unmeasured confounders. Condition A3 means treatment is not allocated deterministically.

Under these three assumptions, it is known that the distribution of either counterfactual may be expressed in terms of observational data. The ultimate goal of this contribution is to model either distribution $Y^a$ using a parametric class of distributions $\mathcal{Q} = \{q_\theta\}_{\theta \in \Theta}$ which only need to be specified up to normalizing constants. Without loss of generality, we restrict our analysis to the potential outcome of the treatment $Y^1 \sim Q^1$. For conducting such a task, we draw upon minimum kernel Stein discrepancy estimators, which build on the concepts of reproducing kernel Hilbert space and Stein's method.

**Reproducing kernel Hilbert spaces (RKHS):** Consider a non-empty set $\mathcal{Y}$ and a Hilbert space $\mathcal{H}$ of functions $f : \mathcal{Y} \to \mathbb{R}$ equipped with the inner product $\langle \cdot, \cdot \rangle_{\mathcal{H}}$. The Hilbert space $\mathcal{H}$ is called an RKHS if there exists a function $k : \mathcal{Y} \times \mathcal{Y} \to \mathbb{R}$ (referred to as *reproducing kernel*) satisfying (i) $k(\cdot, y) \in \mathcal{H}$ for all $y \in \mathcal{Y}$, (ii) $\langle f, k(\cdot, y) \rangle_{\mathcal{H}} = f(y)$ for all $y \in \mathcal{Y}$ and $f \in \mathcal{H}$. We denote by $\mathcal{H}^d$ the product RKHS containing elements $h := (h_1, \ldots, h_d)$ with $h_i \in \mathcal{H}$ and $\langle h, \tilde{h} \rangle = \sum_{i=1}^{d} \langle h_i, \tilde{h}_i \rangle_{\mathcal{H}}$.

**Kernel Stein discrepancies (KSD):** Assume $Q_\theta$ has density $q_\theta$ on $\mathcal{Y} \subset \mathbb{R}^d$, and let $\mathcal{H}$ be an RKHS with reproducing kernel $k$ such that $\nabla k(\cdot, y)$ exists for all $y \in \mathcal{Y}$. Let $s_\theta(y) = \nabla_y \log q_\theta(y)$ and define $\xi_\theta(\cdot, y) := [s_\theta(y)k(\cdot, y) + \nabla k(\cdot, y)] \in \mathcal{H}^d$, so that

$$
\begin{aligned}
h_\theta(y, \tilde{y}) &= \langle \xi_\theta(\cdot, y), \xi_\theta(\cdot, \tilde{y}) \rangle_{\mathcal{H}^d} \\
&= \langle s_\theta(y), s_\theta(\tilde{y}) \rangle_{\mathbb{R}^d} k(y, \tilde{y}) + \langle s_\theta(\tilde{y}), \nabla_y k(y, \tilde{y}) \rangle_{\mathbb{R}^d} + \\
&\quad + \langle s_\theta(y), \nabla_{\tilde{y}} k(y, \tilde{y}) \rangle_{\mathbb{R}^d} + \langle \nabla_y k(\cdot, y), \nabla_{\tilde{y}} k(\cdot, \tilde{y}) \rangle_{\mathcal{H}^d}
\end{aligned}
\tag{1}
$$

is a reproducing kernel. The KSD is defined as $\mathrm{KSD}(Q_\theta \| Q) := (\mathbb{E}_{Y, \tilde{Y} \sim Q}[h_\theta(Y, \tilde{Y})])^{1/2}$. Under certain regularity conditions, $\mathrm{KSD}(Q_\theta \| Q) = 0$ if (and only if) $Q_\theta = Q$ (Chwialkowski et al., 2016); other properties such as weak convergence dominance may also be established (Gorham & Mackey, 2017). Further, if $\mathbb{E}_Q \sqrt{h_\theta(Y, Y)} < \infty$, then $\mathrm{KSD}(Q_\theta \| Q) = \| \mathbb{E}_{Y \sim Q}[\xi_\theta(\cdot, Y)] \|_{\mathcal{H}^d}$.

In non-causal settings, given $Y_1, \ldots, Y_n \sim Q$ and the closed form evaluation of $h_\theta$ presented in equation 1, the V-statistic

$$
V_n(\theta) = \| \frac{1}{n} \sum_{i=1}^{n} \xi_\theta(\cdot, Y_i) \|_{\mathcal{H}^d}^2 = \frac{1}{n^2} \sum_{i=1}^{n} \sum_{j=1}^{n} h_\theta(Y_i, Y_j).
\tag{2}
$$

may be considered for estimating $\mathrm{KSD}^2(Q_\theta \| Q)$. Similarly, removing the diagonal elements of this V-statistic gives way to an unbiased U-statistic, which counts with similar properties.

**Minimum kernel Stein discrepancy (MKSD) estimators:** Given $\mathcal{Q} = \{q_\theta\}_{\theta \in \Theta}$ known up to normalizing constants, the scores $s_\theta$ can be computed. MKSD estimators model $Q$ by $Q_{\theta_n}$, where

$$
\theta_n \in \arg\min_{\theta \in \Theta} g_n(\theta; Y_1, \ldots, Y_n),
$$

and $g_n$ is the V-statistic defined in equation 2 or its unbiased version. The MKSD estimator $\theta_n$ is consistent and asymptotically normal under regularity conditions (Oates et al., 2022).

## 4 MAIN RESULTS

We consider the problem of modeling the counterfactual distribution $Y^1 \sim Q^1$ given a parametric class of distributions $\mathcal{Q} = \{q_\theta\}_{\theta \in \Theta}$ with potentially unknown normalizing constants. We work under the potential outcomes framework, assuming that we have access to observations $(X_i, A_i, Y_i)_{i=1}^{n} \in \mathcal{X} \times \{0, 1\} \times \mathcal{Y}$ sampled as $(X, A, Y) \sim P$, such that Conditions A1-A3 hold. Throughout, we denote $Z \equiv (X, A, Y)$ and $\mathcal{Z} \equiv \mathcal{X} \times \{0, 1\} \times \mathcal{Y}$ for ease of presentation.

Like the previously introduced MKSD estimators, our approach includes modeling $Q^1$ by choosing a $\theta$ such that

$$
\theta_n \in \arg\min_{\theta \in \Theta} g_n(\theta; Z_1, \ldots, Z_n),
\tag{3}
$$

where $g_n$ is a proxy for $\mathrm{KSD}^2(Q_\theta \| Q^1)$. In contrast to such MKSD estimators, defining $g_n$ as the V-statistic introduced in equation 2 would lead to inconsistent estimations, given that the distribution of $Y$ may very likely differ from that of $Y^1$ (this is, indeed, the very essence of counterfactual settings).

In order to define an appropriate $g_n$, we first note that, by the law of iterated expectation,

$$\Psi := \mathbb{E}_{Y^1 \sim Q^1}\left[\xi_\theta(\cdot, Y^1)\right] = \mathbb{E}_{Z \sim P}\left[\phi(Z)\right],$$

where

$$\phi(z) = \frac{a}{\pi(x)}\left\{\xi_\theta(\cdot, y) - \beta_\theta(x)\right\} + \beta_\theta(x), \tag{4}$$
$$\pi(x) = \mathbb{E}\left[A | X = x\right], \quad \beta_\theta(x) = \mathbb{E}\left[\xi_\theta(\cdot, Y) | A = 1, X = x\right].$$

An analogous result holds for $Y^0$ or any other discrete treatment level. Note that the $Y^0$ does not affect neither $\pi$ nor $\beta_\theta$. In fact, the problem could have been presented as a missing outcome data problem, where the data from $Y^0$ is missing. The authors have posed the problem in counterfactual outcome terms simply for motivational purposes.

The embedding $\phi$ induces a reproducing kernel $h_\theta^*(z, \tilde{z}) = \langle \phi(z), \phi(\tilde{z}) \rangle_{\mathcal{H}^d}$, which will be used in subsequent theoretical analyses. We highlight that $\phi - \Psi$ is the *efficient influence function* for $\Psi$. This implies that the resulting estimators have desired statistical properties such as double robustness, as shown later. That is, $\hat{\Psi} = P_n \hat{\phi}_\theta$ will be consistent if either $\hat{\pi}$ or $\hat{\beta}_\theta$ is consistent, and its convergence rate corresponds to the product of the learning rates of $\hat{\pi}$ and $\hat{\beta}_\theta$.

Nonetheless, it may not be feasible to estimate $\hat{\beta}_\theta$ individually for each $\theta$ of interest. In turn, let us assume for now that we have access to estimators $\hat{\pi}$ and $\hat{\beta}$, where the latter approximates $\beta(x) = \mathbb{E}\left[k(\cdot, Y) | A = 1, X = x\right]$, and is of the form

$$\hat{\beta}(x) = \sum_{i=1}^{n} \hat{w}_i(x) k(\cdot, Y_i). \tag{5}$$

We highlight that many prominent algorithms for conducting $\mathcal{H}^d$-valued regression, such as conditional mean embeddings (Song et al., 2009; Grünewälder et al., 2012) and distributional random forests (Ćevid et al., 2022; Näf et al., 2023), are of the form exhibited in equation 5. In order to avoid refitting the estimator $\hat{\beta}_\theta$ for each value of $\theta$, we propose to define $\hat{\beta}_\theta$ from $\hat{\beta}$ as follows:

$$\hat{\beta}_\theta(x) = [\hat{\beta}_\theta(\hat{\beta})](x) := \sum_{i=1}^{n} \hat{w}_i(x) \xi_\theta(\cdot, Y_i). \tag{6}$$

Intuitively, we could expect that if estimator $\hat{\beta}$ is consistent, then $\hat{\beta}_\theta$ is also consistent if the mapping $k(\cdot, y) \to [s_\theta(y) k(\cdot, y) + \nabla k(\cdot, y)]$ is regular enough. As a result, we propose the statistic

$$g_n(\theta; Y_1, \ldots, Y_n) = \left\langle \frac{1}{n} \sum_{i=1}^{n} \hat{\phi}_\theta(Z_i), \frac{1}{n} \sum_{j=1}^{n} \hat{\phi}_\theta(Z_j) \right\rangle_{\mathcal{H}^d}, \tag{7}$$

where $\hat{\beta}_\theta$ is constructed from $\hat{\beta}$ as established in equation 6. Although this statistic resembles the one presented in Martinez-Taboada et al. (2023) and Fawkes et al. (2022), several important differences arise between the contributions. First and foremost, they studied a testing problem, so the causal target was different. Second, their work did not extend to embeddings that depend on a parameter $\theta$. Lastly, they focused on the distribution of their statistic under the null in order to calibrate a two-sample test; in contrast, our approach minimizes a related statistic for directly modelling the counterfactual. The main theoretical contribution of Martinez-Taboada et al. (2023) is the extension of cross U-statistics to the causal setting; in stark contrast, our contribution deals with the theoretical properties of the minimizer of a 'debiased' V-statistic. Both the motivation and theoretical challenges in our study diverge from those in the earlier works; we view our research as complementary and orthogonal to these prior contributions.

While we have assumed so far that estimators $\hat{\pi}$ and $\hat{\beta}_\theta$ are given, defining an optimal strategy for training such estimators is key when seeking to maximize the performance. A simple approach, such as using half of the data to estimate $\pi$ and $\beta_\theta$, and the other half on the empirical averages of the

---

**Algorithm 1** DR-MKSD

1: **input** Data $\mathcal{D} = (X_i, A_i, Y_i)_{i=1}^n$ and $\mathcal{Q} = \{q_\theta\}_{\theta \in \Theta}$ such that that $s_\theta$ is known for all $\theta$ in $\Theta$.
2: **output** A distribution $q_{\theta_n}$ that approximates the potential outcome $Y^1$.
3: Choose kernel $k$ and estimators $\hat{\beta}$, $\hat{\pi}$.
4: Split data in two sets $\mathcal{D}_1 = (X_i, A_i, Y_i)_{i=1}^{n//2}$ and $\mathcal{D}_2 = (X_i, A_i, Y_i)_{i=n//2+1}^n$.
5: Train $\hat{\pi}^{(1)}$, $\hat{\beta}^{(1)}$ on $\mathcal{D}_2$ and $\hat{\pi}^{(2)}$, $\hat{\beta}^{(2)}$ on $\mathcal{D}_1$. Define $\hat{\beta}_\theta^{(r)}$ from $\hat{\beta}^{(r)}$ as in equation 6.
6: Define $\hat{\phi}_\theta$ as $\hat{\phi}_\theta^{(r)}$ on $\mathcal{D}_{1-r}$ for $r \in \{1, 2\}$, and $g_n$ as in equation 7.
7: Take $\theta_n \in \arg\min_{\theta \in \Theta} g_n(\theta; Y_1, \ldots, Y_n)$.
8: **return** $q_{\theta_n}$.

---

statistic $g_n$, would lead to an increase in the variance of the latter. So, we use cross-fitting (Robins et al., 2008; Zheng & van der Laan, 2010; Chernozhukov et al., 2018). This is, split the data in half, use the two folds to train different estimators separately, and then evaluate each estimator on the fold that was not used to train it.

Based on these theoretical considerations, we propose a novel estimator called DR-MKSD (Doubly Robust Minimum Kernel Stein Discrepancy) outlined in Algorithm 1. Two key observations regarding Algorithm 1 are noteworthy. Although we have presented $g_n$ as an abstract inner product in $\mathcal{H}^d$, equation 7 has a closed-form expression as long as $h_\theta$ can be evaluated. Further details can be found in Appendix A. Additionally, the estimator $\theta_n$ is defined through a minimization problem, the complexity of which depends on $E_\theta$. In general, $g_n$ need not be convex with respect to $\theta$, and thus, estimating $\theta_n$ may involve typical non-convex optimization challenges. However, this is inherent to our approach, aiming for flexible modeling of the counterfactual distribution. The potential $E_\theta$ itself could be a neural network, making non-convexity an unavoidable aspect of the problem.

We note that evaluating $g_n$ has a time complexity of $O(n^2)$. Nonetheless, in stark contrast to the methods presented in Kim et al. (2018); Kennedy et al. (2021), the proposed DR-MKSD procedure only requires the nuisance estimators to be fitted once. This enables DR-MKSD to leverage computationally expensive estimators, such as deep neural networks, providing a significant advantage with respect to these previous contributions.

While it may difficult to find a global minimizer of $g_n$, we turn to study the properties of the estimator as if that optimization task posed no problem, with the scope that this analysis sheds light on the expected behaviour of the procedure if the minimization problem yields a good enough estimate of $\theta$. We thus provide sufficient conditions for consistency and inference properties of the optimal $\theta_n$. We defer the proofs to Appendix C, as well as an exhaustive description of the notation used.

**Theorem 1 (Consistency)** *Assume that $\Theta \subset \mathbb{R}^p$ open, convex, and bounded. Further, let Conditions A1-A3 hold, as well as*

$$A4) \iint \sup_{\theta \in \Theta} h_\theta^*(z, \tilde{z}) dP(z) dP(\tilde{z}) < \infty, \qquad A5) \int \sup_{\theta \in \Theta} h_\theta^*(z, z) dP(z) < \infty,$$

$$A6) \iint \sup_{\theta \in \Theta} \|\partial_\theta h_\theta^*(z, \tilde{z})\|_{\mathbb{R}^p} dP(z) dP(\tilde{z}) < \infty, \qquad A7) \int \sup_{\theta \in \Theta} \|\partial_\theta h_\theta^*(z, z)\|_{\mathbb{R}^p} dP(z) < \infty.$$

*If (i) $P(\hat{\pi}^{(r)} \in [\epsilon, 1-\epsilon]) = 1$, (ii) $\sup_{\theta \in \Theta} \|\hat{\phi}_\theta^{(r)} - \phi_\theta\| = o_P(\sqrt{n})$ and (iii) $\|\hat{\pi}^{(r)} - \pi\| \sup_{\theta \in \Theta} \|\hat{\beta}_\theta^{(r)} - \beta_\theta\| = o_P(1)$ for $r \in \{1, 2\}$, then*

$$KSD(Q_{\theta_n} \| Q^1) \xrightarrow{p} \min_{\theta \in \Theta} KSD(Q_\theta \| Q^1).$$

Conditions A4-A7 build on the supremum of an abstract kernel $h_\theta^*$, but we note that the properties stem from those of $k$ and $s_\theta$. If $s_\theta(y)$, $\partial_\theta s_\theta(y)$, $k(\cdot, y)$, and $\nabla_y k(\cdot, y)$ are bounded, then so are $h_\theta^*$ and $\partial_\theta h_\theta^*$, and Conditions A4-A7 are fulfilled. In particular, if $s_\theta$ and $k$ are smooth and $\mathcal{Y}$ is compact, such assumptions are attained. We highlight the weakness of Condition (i), which only requires that our estimates are bounded away from 0 and 1, and Condition (ii), which does not even require the estimates of $\phi_\theta$ to be consistent, and is usually implied by Condition (iii).

Condition (iii) implicitly conveys the so-called double robustness. That is, as long the estimator of $\pi$ is consistent or the estimators of $\beta_\theta$ are uniformly consistent, then so will be the procedure. We

highlight that we are not estimating $\beta_\theta$ independently for each $\theta$, but we are rather constructing all of them based on an estimate of $\beta$. This opens the door to having immediate uniform consistency as long as the estimator of $\beta$ is consistent (depending on the underlying structure between $\beta$ and $\beta_\theta$) in a similar spirit to the work on uniform consistency rates presented in Hardle et al. (1988).

The consistency properties of a doubly robust estimator are highly desirable. However, a most important characteristic is the convergence rate, which we show next corresponds to the product of the convergence rates of the two estimators upon which it is constructed.

**Theorem 2 (Asymptotic normality)** *Let $\Theta \subset \mathbb{R}^p$ be open, convex, and bounded, and assume $\theta_n \xrightarrow{p} \theta_*$, where $KSD(Q_{\theta_*}\|Q^1) = \min KSD(Q_\theta\|Q^1)$. Suppose Conditions A1-A5 hold, as well as A8) the maps $\{\theta \mapsto \partial_\theta h_\theta^*(z, \tilde{z}) : z, \tilde{z} \in \mathcal{Z}\}$ are differentiable on $\Theta$, A9) the maps $\{\theta \mapsto \partial_\theta^2 h_\theta^*(z, \tilde{z}) : z, \tilde{z} \in \mathcal{Z}\}$ are uniformly continuous at $\theta_*$, A10) $\iint \sup_{\theta \in \Theta} \|\partial_\theta h_\theta^*(z, \tilde{z})\|_{\mathbb{R}^p} dP(z) dP(\tilde{z}) < \infty$, A11) $\iint \|\partial_\theta h_\theta^*(z, \tilde{z})\|_{\mathbb{R}^p}^2 dP(z) dP(\tilde{z}) \mid_{\theta=\theta_*} < \infty$, A12) $\int \sup_{\theta \in \Theta} \|\partial_\theta^2 h_\theta^*(z, z)\|_{\mathbb{R}^{p \times p}} dP(z) < \infty$, A13) $\iint \sup_{\theta \in \Theta} \|\partial_\theta^2 h_\theta^*(z, \tilde{z})\|_{\mathbb{R}^{p \times p}} dP(z) dP(\tilde{z}) < \infty$, A14) $\Gamma = \iint \partial_\theta^2 h_\theta^*(z, \tilde{z}) dP(z) dP(\tilde{z}) \mid_{\theta=\theta_*} \succ 0$.*

*If (i') $P(\hat{\pi}^{(r)} \in [\epsilon, 1 - \epsilon]) = 1$, (ii') $\sup_{\theta \in \Theta} \left\{ \|\hat{\phi}_\theta^{(r)} - \phi_\theta\| + \|\partial_\theta \hat{\phi}_\theta^{(r)} - \partial_\theta \phi_\theta\| \right\} = o_P(1)$, and*

$$(iii') \ \|\hat{\pi}^{(r)} - \pi\| \sup_{\theta \in \Theta} \left\{ \|\hat{\beta}_\theta^{(r)} - \beta_\theta\| + \|\partial_\theta \hat{\beta}_\theta^{(r)} - \partial_\theta \beta_\theta\| \right\} = o_P(\frac{1}{\sqrt{n}})$$

*for $r \in \{1, 2\}$, then*

$$\sqrt{n}(\theta_n - \theta^*) \xrightarrow{d} N(0, 4\Gamma^{-1}\Sigma\Gamma^{-1}),$$

*where $\Sigma := Cov_{Z \sim P}[\int \partial_\theta h_\theta^*(Z, \tilde{z}) dP(\tilde{z})] \mid_{\theta=\theta_*}$.*

As proposed in Oates et al. (2022), a sandwich estimator $4\Gamma_n^{-1}\Sigma_n\Gamma_n^{-1}$ for the asymptotic variance $4\Gamma^{-1}\Sigma\Gamma-1$ can be established (Freedman, 2006), where $\Sigma_n$ and $\Gamma_n$ are obtained by substituting $\theta_*$ by $\theta_n$ and replacing the expectations by empirical averages. If the estimator $4\Gamma_n^{-1}\Sigma_n\Gamma_n^{-1}$ is consistent at any rate, then $\Sigma_n^{-1/2}\Gamma_n(\theta_* - \theta_n) \xrightarrow{d} N(0, 1)$ in view of Slutsky's theorem.

Conditions A8-A13 can be satisfied under diverse regularity assumptions, analogously to Conditions A4-A7. Further, we note that assumptions akin to Condition A14 are commonly encountered in the context of asymptotic normality. While Condition (i') is carried over the consistency analysis, Condition (ii') now requires the estimators of $\phi_\theta$ and $\partial_\theta\phi_\theta$ to be uniformly consistent. However, we highlight the weakness of this assumption. For instance, it is attained as long as $\hat{\pi}$ is consistent and $\hat{\beta}_\theta$ is uniformly bounded. Lastly, we again put the spotlight on Condition (iii'): if the product of the rates is $o_P(n^{-1/2})$, asymptotic normality of the estimate $\theta_n$ can be established.

The double robustness of $g_n$ has two profound implications in the DR-MKSD procedure. First, $g_n(\theta)$ converges to $KSD^2(Q_\theta\|Q^1)$ faster than if solely relying on nuisance estimators $\hat{\pi}$ or $\hat{\beta}_\theta$, which translates to a better estimate $\theta_n$. Further, it opens the door to conducting inference if the product of the rates is $o_P(n^{-1/2})$, which may be achieved by a rich family of estimators.

We underscore that $\theta_n \xrightarrow{p} \theta_*$ need not be a consequence of Theorem 1. Theorem 1 establishes consistency on $KSD(Q_{\theta_n}\|Q^1)$, not $\theta_n$ itself. Consistent estimators for $KSD(Q_{\theta_n}\|Q^1)$ but not for $\theta_n$ may arise, for example, if the minimizer is not unique. In such a case, one cannot theoretically derive the consistency of the estimate (at least, while remaining agnostic about the optimization solver). For instance, if there are two minimizers and the optimization solver alternates between the two of them, $KSD(Q_{\theta_n}\|Q^1)$ is consistent but $\theta_n$ does not even converge. Nonetheless, we emphasize that the theorems provide sufficient, but not necessary, conditions for consistency and asymptotic normality.

Furthermore, a potential lack of consistency does not necessarily translate to a poor performance of the proposed estimator. Two distributions characterized by very different parameters $\theta$ may be close in the space of distributions defined by the KSD metric (analogously to two neural networks with very different weights having similar empirical performance). Estimating the distribution characterized via the parameter, not the parameter itself, is the ultimate goal of this work. Hence, the proposed estimator can yield good models for the counterfactual distribution even with inconsistent $\theta_n$.

Lastly, we highlight that the DR-MKSD procedure can be easily redefined by estimating $\beta_{\theta_g}$ for each $\theta_g$ belonging to a finite grid $\{\theta_g\}_{g \in G}$. The uniform consistency assumptions would consequently translate on usual consistency for each $\theta_g$ of the set $\{\theta_g\}_{g \in G}$, and the minimization problem would reduce to take the $\arg\min$ of a finite set. The statistical and computational trade-off is clear: estimating $\beta_{\theta_g}$ for each $\theta_g$ may lead to stronger theoretical guarantees and improved performance, but the computational cost of estimating $\beta_{\theta_g}$ increases by a factor of $|G|$.

## 5 EXPERIMENTS

We provide a number of experiments with (semi)synthetic data. Given that this is the first estimator to handle unnormalized densities in the counterfactual setting, we found no natural benchmark to compare the empirical performance of the proposed estimator with. Hence, we focus on illustrating the theoretical properties and exploring the empirical performance of the estimator in a variety of settings. Throughout, we take $k(x, y) = (c^2 + l^{-2}\|x - y\|_{\mathbb{R}^d}^2)^\beta$ to be the inverse multi-quadratic (IMQ) kernel, based on the discussion in Gorham & Mackey (2017), with $\beta = -0.5$, $l = 0.1$ and $c = 1$. We estimate the minimizer of $g_n(\theta)$ by gradient descent. We defer an exhaustive description of all simulations and further experiments to Appendix B.

**Consistency of the DR-MKSD estimator:** To elucidate the double robustness of DR-MKSD, we define its inverse probability weighting and plug-in versions, given by $\hat{\phi}_{\text{IPW},\theta}(z) = \frac{a}{\hat{\pi}(x)}\xi_\theta(\cdot, y)$ and $\hat{\phi}_{\text{PI},\theta}(z) = \hat{\beta}_\theta(x)$ respectively. We draw $(X_i, A_i, Y_i)_{i=1}^n$ such that $Y^1 \sim \mathcal{N}(0, 1)$. Further, we let $X \sim \mathcal{N}(Y^1, 1)$ and we sample $A$ using a logistic model that depends on $X$. Note that this sampling procedure is consistent with Conditions A1-A3. We consider the set of normal distributions with unit variance $\mathcal{Q} = \{\mathcal{N}(\theta, 1)\}_{\theta \in \mathbb{R}}$. Figure 1 exhibits the mean squared error of the procedures across 100 bootstrap samples, different sample sizes $n$ and various choices of estimators. The procedure shows consistency as long as either the IPW or PI versions are consistent, even if the other is not. We defer to Appendix B an analysis of the confidence intervals for $\theta_n$ given by Theorem 2, where we illustrate that they have the desired empirical coverage.

**Asymptotic normality of the DR-MKSD estimator:** Following the examples in Liu et al. (2019) and Matsubara et al. (2022), we consider the intractable model with potential function $-E_\theta(y) = \langle \eta(\theta), J(y) \rangle_{\mathbb{R}^s}$, where $\theta \in \mathbb{R}^2$, $y \in \mathbb{R}^5$, $\eta(\theta) := (-0.5, 0.6, 0.2, 0, 0, 0, \theta)^T$, and $J(y) = (\sum_{i=1}^5 y_i^2, y_1 y_2, \sum_{i=3}^5 y_1 y_i, tanh(y))^T$. The normalizing constant is not tractable except for $\theta = 0$, where we recover the density of a Gaussian distribution $\mathcal{N}_0$. We draw $(X_i, A_i, Y_i)_{i=1}^{500}$ so that $Y^1 \sim \mathcal{N}_0$ and $X \sim \mathcal{N}(Y^1, I_5)$. Lastly, $A$ follows a logistic model that depends on $X \odot X$. Figure 2 displays the empirical estimates of $\theta = (\theta_1, \theta_2)$, which shows approximately normal.

**Counterfactual Restricted Boltzmann Machines (RBM):** We let $(X_i, A_i, Y_i)_{i=1}^{500}$ such that $Y^1$ is drawn from a two-dimensional RBM with one hidden variable $h$ such that $q_\theta(y^1, h) \propto \exp(h + \langle \theta, y^1 \rangle_{\mathbb{R}^2} - 2\|y^1\|_2^2)$. Additionally, $X \sim \mathcal{N}(Y^1, 0.25I_2)$ and $A$ follows a logistic model that depends

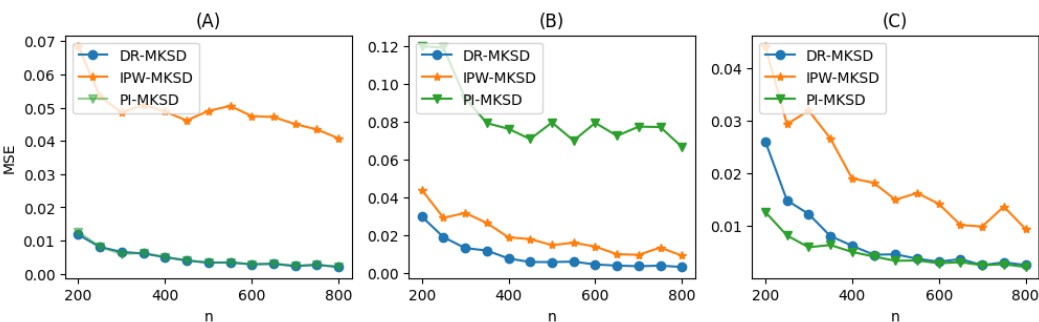

Figure 1: Mean squared error (MSE) of $\theta_n$ approximated with 100 bootstrap samples. $\pi$ and $\beta$ are estimated via (A) boosting and conditional mean embeddings (CME), (B) logistic regression and 1-NN, (C) logistic regression and CME. The doubly robust approach proves consistent in all cases.

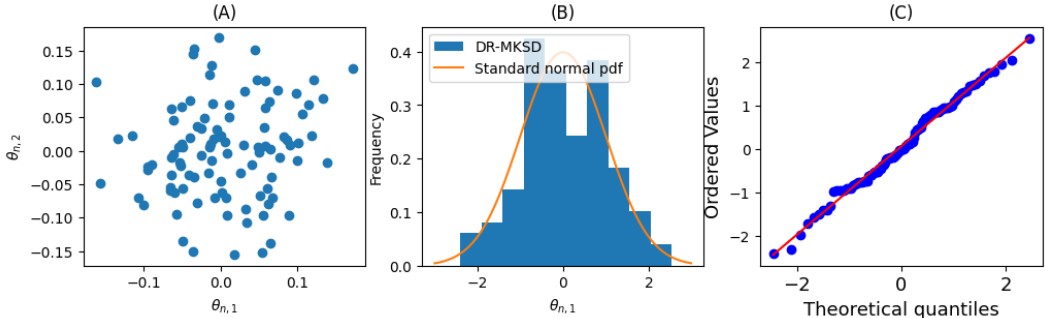

Figure 2: Illustration of 100 simulations of the DR-MKSD estimator $\theta_n = (\theta_{n,1}, \theta_{n,2})$, constructed on estimates of $\pi$ and $\beta$ fitted as random forests and CME for $n = 500$, with (A) scatter plot of empirical distribution for $\theta_n$, (B) histogram of the standardized values of $\theta_{n,1}$, (C) Normal QQ plot of the standardized values of $\theta_{n,1}$. We highlight the two-dimensional Gaussian behaviour of $\theta_n$.

on $X$. Although estimating the exact $\theta$ is hopeless ($h$ is unobservable), it may still be possible to uncover the underlying structure governing the behavior of $Y^1$, which depends on the ratio between $\theta_1$ and $\theta_2$. Figure 3 shows that minimizing $g_n$ does indeed recover the direction of $\theta$.

**Experimental data:** Suppose one has access to a noisy version of $Y^1$, denoted as $X$, and the process of denoising this data to recover $Y^1$ is costly. Due to budget constraints, one can only afford to denoise approximately half the data. The objective is to estimate the distribution of $Y^1 \in \mathbb{R}^d$, which can be described by a model $q_\theta(Y^1) \propto \exp(\langle T(Y^1), \theta \rangle)$. Here, $T(Y^1) \in \mathbb{R}^p$ represents a lower-dimensional representation of $Y^1$. To achieve this, one randomly selects observations for denoising with a probability of 1/2 and employs DR-MKSD to model $Y^1$.

For illustration, we consider ten distinct counterfactuals, denoted as $Y^{1,i} \in \mathbb{R}^{20}$ for $i \in \{1, \ldots, 10\}$. These counterfactuals are defined as $Y^{1,i} := f(W^i)$, where each $W^i$ represents an observation from the MNIST dataset corresponding to digit $i$, and $f$ is a pretrained neural network. Additionally, we utilize another pretrained neural network $T : \mathbb{R}^{20} \to \mathbb{R}^{10}$. In Figure 4, we display random MNIST dataset observations in the top row, alongside those with the lowest unnormalized density of $f(W^i)$ in the bottom row, estimated by DR-MKSD independently for each digit. Notably, the latter images appear more distorted, which aligns with the expectation that DR-MKSD accurately models $Y^1$.

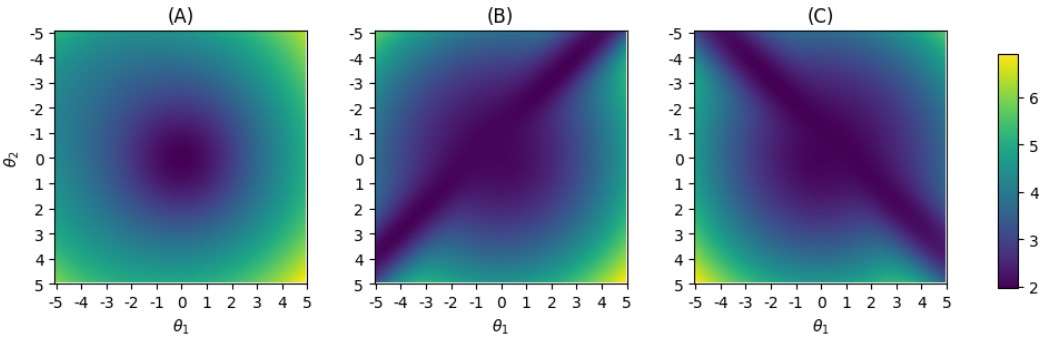

Figure 3: Values of the statistic $g_n(\theta)$ for the RBM with (A) $\theta_1 = \theta_2 = 0$, (B) $\theta_1 = \theta_2 = 1$, (C) $\theta_1 = -\theta_2 = 1$. Minimizing $g_n$ leads to recover the direction of $\theta$.

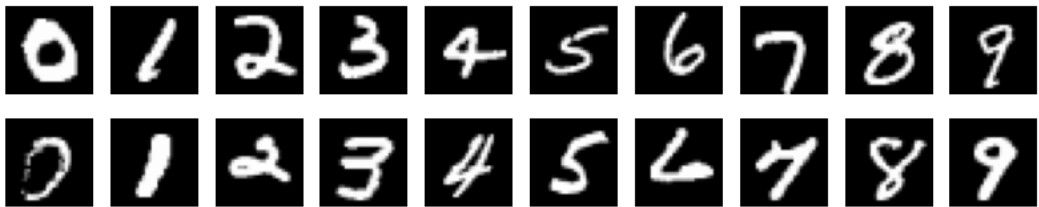

Figure 4: Observations of the MNIST dataset chosen randomly (top row), and with the lowest estimated density of $f(W^i)$ (bottom row). We highlight the potential outlier detection in digits 0, 2, 4 and 6.

## 6 DISCUSSION

We have presented an estimator for modeling counterfactual distributions given a flexible set of distributions, which only need to be known up to normalizing constants. The procedure builds on minimizing the kernel Stein discrepancy between such a set and the counterfactual, while simultaneously accounting for sampling bias in a doubly robust manner. We have provided sufficient conditions for the consistency and asymptotic normality of the estimator, and we have illustrated its performance in various scenarios, showing the empirical validity of the procedure.

There are several avenues for future research. Employing energy-based models for estimating the counterfactual enables the generation of synthetic observations using a rich representation of the potential outcome. Exploring the empirical performance of sampling methods that do not require the normalizing constants, such as Hamiltonian Monte Carlo or the Metropolis-Hasting algorithm, holds particular promise in domains where collecting additional real-world data is challenging.

Furthermore, extensions could involve incorporating instrumental variables and conditional effects. Of particular interest would be the expansion of our framework to accommodate time-varying treatments. Lastly, we highlight that, while we have framed the problem within a causal inference context, analogous scenarios arise in off-policy evaluation (Dudík et al., 2011; Thomas & Brunskill, 2016). Extending our contributions to that domain could yield intriguing insights.

### REPRODUCIBILITY STATEMENT

Reproducible code for all experiments is provided in the supplementary materials.

### ACKNOWLEDGMENTS

DMT gratefully acknowledges that the project that gave rise to these results received the support of a fellowship from 'la Caixa' Foundation (ID 100010434). The fellowship code is LCF/BQ/EU22/11930075. EK was supported by NSF CAREER Award 2047444.

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

# A CLOSED FORM OF THE STATISTIC

Under estimators of the form $\hat{\beta}_\theta(x) = \sum_{i=1}^n \hat{w}_i(x)\xi_\theta(\cdot, Y_i)$, we derive that both inner products

$$
\begin{aligned}
\left\langle \hat{\beta}_\theta(x), \hat{\beta}_\theta(\tilde{x}) \right\rangle_{\mathcal{H}^d} &= \left\langle \sum_{i=1}^n \hat{w}_i(x)\xi_\theta(\cdot, Y_i), \sum_{j=1}^n \hat{w}_j(\tilde{x})\xi_\theta(\cdot, Y_j) \right\rangle_{\mathcal{H}^d} \\
&= \sum_{i=1}^n \sum_{j=1}^n \hat{w}_i(x)\hat{w}_j(\tilde{x}) \left\langle \xi_\theta(\cdot, Y_i), \xi_\theta(\cdot, Y_j) \right\rangle_{\mathcal{H}^d} \\
&= \sum_{i=1}^n \sum_{j=1}^n \hat{w}_i(x)\hat{w}_j(\tilde{x}) h_\theta(Y_i, Y_j),
\end{aligned}
$$

$$
\begin{aligned}
\left\langle \hat{\beta}_\theta(x), \xi_\theta(\cdot, Y) \right\rangle_{\mathcal{H}^d} &= \left\langle \sum_{i=1}^n \hat{w}_i(x)\xi_\theta(\cdot, Y_i), \xi_\theta(\cdot, Y) \right\rangle_{\mathcal{H}^d} \\
&= \sum_{i=1}^n \hat{w}_i(x) \left\langle \xi_\theta(\cdot, Y_i), \xi_\theta(\cdot, Y) \right\rangle_{\mathcal{H}^d} \\
&= \sum_{i=1}^n \hat{w}_i(x) h_\theta(Y_i, Y)
\end{aligned}
$$

count with closed forms as long as $h_\theta$ can be computed. Consequently

$$
\begin{aligned}
g_n(\theta; Y_1, \ldots, Y_n) &= \left\langle \frac{1}{n} \sum_{i=1}^n \hat{\phi}_\theta(Z_i), \frac{1}{n} \sum_{j=1}^n \hat{\phi}_\theta(Z_j) \right\rangle_{\mathcal{H}^d} \\
&= \left\langle \frac{1}{n} \sum_{i=1}^n \frac{A_i}{\hat{\pi}(X_i)} \left\{ \xi_\theta(\cdot, Y_i) - \hat{\beta}_\theta(X_i) \right\} + \hat{\beta}_\theta(X_i), \right. \\
&\qquad \left. \frac{1}{n} \sum_{j=1}^n \frac{A_i}{\hat{\pi}(X_j)} \left\{ \xi_\theta(\cdot, Y_j) - \hat{\beta}_\theta(X_j) \right\} + \hat{\beta}_\theta(X_j) \right\rangle_{\mathcal{H}^d} \\
&= \left\langle \frac{1}{n} \sum_{i=1}^n \frac{A_i}{\hat{\pi}(X_i)} \xi_\theta(\cdot, Y_i) + \left\{ 1 - \frac{A_i}{\hat{\pi}(X_i)} \right\} \hat{\beta}_\theta(X_i), \right. \\
&\qquad \left. \frac{1}{n} \sum_{j=1}^n \frac{A_j}{\hat{\pi}(X_j)} \xi_\theta(\cdot, Y_j) + \left\{ 1 - \frac{A_j}{\hat{\pi}(X_j)} \right\} \hat{\beta}_\theta(X_j) \right\rangle_{\mathcal{H}^d} \\
&= \frac{1}{n^2} \sum_{i=1}^n \sum_{j=1}^n \frac{A_i A_j}{\hat{\pi}(X_i)\hat{\pi}(X_j)} \left\langle \xi_\theta(\cdot, Y_i), \xi_\theta(\cdot, Y_j) \right\rangle_{\mathcal{H}^d} + \\
&\quad + \frac{2}{n^2} \sum_{i=1}^n \sum_{j=1}^n \frac{A_i}{\hat{\pi}(X_i)} \left\{ 1 - \frac{A_j}{\hat{\pi}(X_j)} \right\} \left\langle \xi_\theta(\cdot, Y_i), \hat{\beta}_\theta(X_j) \right\rangle + \\
&\quad + \frac{1}{n^2} \sum_{i=1}^n \sum_{j=1}^n \left\{ 1 - \frac{A_i}{\hat{\pi}(X_i)} \right\} \left\{ 1 - \frac{A_j}{\hat{\pi}(X_j)} \right\} \left\langle \hat{\beta}_\theta(X_i), \hat{\beta}_\theta(X_j) \right\rangle \\
&= \frac{1}{n^2} \sum_{i,j=1}^n \frac{A_i A_j}{\hat{\pi}(X_i)\hat{\pi}(X_j)} h_\theta(Y_i, Y_j) + \\
&\quad + \frac{2}{n^2} \sum_{i,j,i'=1}^n \frac{A_i}{\hat{\pi}(X_i)} \left\{ 1 - \frac{A_j}{\hat{\pi}(X_j)} \right\} \hat{w}_{i'}(X_j) h_\theta(Y_{i'}, Y_i) + \\
&\quad + \frac{1}{n^2} \sum_{i,j,i',j'=1}^n \left\{ 1 - \frac{A_i}{\hat{\pi}(X_i)} \right\} \left\{ 1 - \frac{A_j}{\hat{\pi}(X_j)} \right\} \hat{w}_{i'}(X_i)\hat{w}_{j'}(X_j) h_\theta(Y_{i'}, Y_{j'})
\end{aligned}
$$

is closed form as long as $h_\theta$, $\hat{\pi}$ and $w$ can be evaluated.

## B EXPERIMENTS

### B.1 CONSISTENCY OF THE DR-MKSD ESTIMATOR

We consider the set of normal distributions with unit variance $\mathcal{Q} = \{\mathcal{N}(\theta, 1)\}_{\theta \in \mathbb{R}}$. We draw $(X_i, A_i, Y_i)_{i=1}^n$ for $n \in \{200, 250, \ldots, 800\}$ such that $Y^1 \sim \mathcal{N}(0, 1)$. Further, $X \sim \mathcal{N}(Y^1, 1)$ and $A$ follows a Bernoulli distribution with log odds $X$.

For obtaining Figure 1, we estimate $\pi$ with

- Default *LogisticRegression* from the *scikit-learn* package with *C* = 1e5 and *max_iter* = 1000,

- Default *AdaBoostClassifier* from the *scikit-learn* package,

and $\beta$ with

- Conditional Mean Embeddings (CME) with the radial basis function kernel and regularization parameter $1e - 3$,

- Vector-valued One Nearest Neighbor (1-NN).

Parameter $\theta$ was estimated by gradient descent for a number of 1000 steps. A total number of 100 experiments with different random seeds were run in order to yield Figure 1.

We further analyze the empirical coverage of the estimated 0.95-confidence intervals $\theta_n \pm \Delta_n$, where $\Delta_n = 1.96\sqrt{4\Gamma_n^{-1}\Sigma_n\Gamma_n^{-1}/n}$ and $\Sigma_n$ and $\Gamma_n$ are defined as described in Section 4. Figure 5 exhibits the confidence intervals for $n = 200$ and $n = 300$. We highlight that the respective empirical coverage for those sample sizes are $0.94$ and $0.96$ respectively, hence presenting a desired empirical coverage.

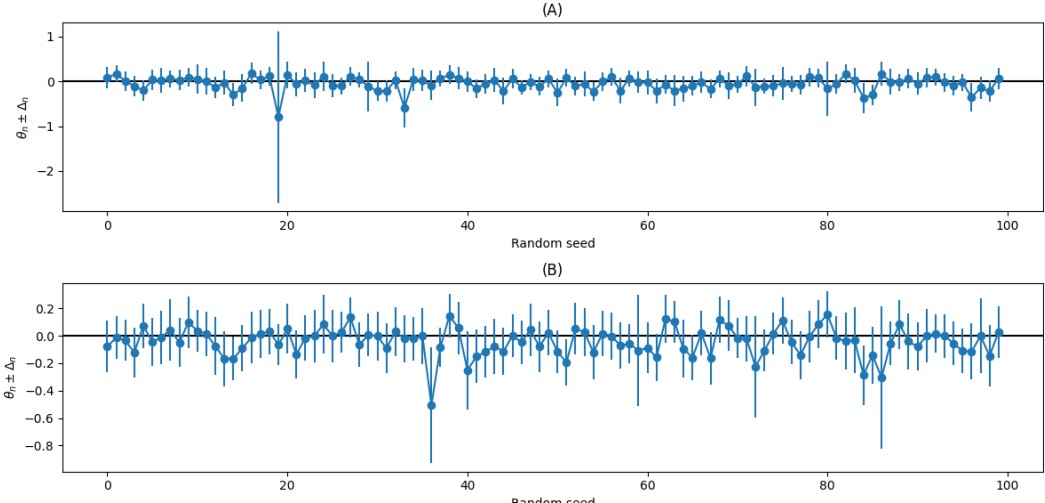

Figure 5: Estimated 0.95-confidence intervals $\theta_n \pm \Delta_n$, where $\Delta_n = 1.96 * \sqrt{4\Gamma_n^{-1}\Sigma_n\Gamma_n^{-1}/n}$ for (A) $n = 200$, and (B) $n = 300$. The empirical coverages of the confidence intervals are (A) 0.94, and (B) 0.96.

## B.2 Asymptotic normality of the DR-MKSD estimator

We consider the intractable model $q_\theta(y) \propto \exp(-E_\theta(y))$ with potential function $-E_\theta(y) = \langle \eta(\theta), J(y) \rangle_{\mathbb{R}^8}$, where

$$\eta(\theta) := (-0.5, 0.6, 0.2, 0, 0, 0, \theta)^T, \quad \theta \in \mathbb{R}^2,$$

$$J(y) := (\sum_{i=1}^{5} y_i^2, y_1 y_2, \sum_{i=3}^{5} y_1 y_i, tanh(y))^T, \quad y \in \mathbb{R}^5.$$

The normalizing constant is not tractable except for $\theta = 0$, where we recover the density of a Gaussian distribution $\mathcal{N}(0, \Sigma)$, with

$$\Sigma^{-1} = \begin{pmatrix} 1 & -0.6 & -0.2 & -0.2 & -0.2 \\ -0.6 & 1 & 0 & 0 & 0 \\ -0.2 & 0 & 1 & 0 & 0 \\ -0.2 & 0 & 0 & 1 & 0 \\ -0.2 & 0 & 0 & 0 & 1 \end{pmatrix}.$$

We draw $(X_i, A_i, Y_i)_{i=1}^{500}$ so that $Y^1 \sim \mathcal{N}(0, \Sigma)$ and $X \sim \mathcal{N}(Y^1, I_5)$. Lastly, $A$ follows a Bernoulli distribution with log odds $\sum_{i=1}^{5}(X_i^2 - 1)$.

For obtaining Figure 2, we estimate $\pi$ with the default *RandomForestClassifier* from the *scikit-learn* package, and $\beta$ with Conditional Mean Embeddings (CME) with the radial basis function kernel and regularization parameter $1e-3$. Parameter $\theta$ was estimated by gradient descent for a number of 1000 steps. A total number of 100 experiments with different random seeds were run.

## B.3 Counterfactual Restricted Boltzmann Machines (RBM)

We let $(X_i, A_i, Y_i)_{i=1}^{500}$ such that $Y^1$ is drawn from a two-dimensional RBM with one hidden variable $h$ such that $q_\theta(y^1, h) \propto \exp(h + \langle \theta, y^1 \rangle_{\mathbb{R}^2} - 2\|y^1\|_2^2)$, for $\theta = (0, 0)$, $\theta = (1, 1)$, and $\theta = (1, -1)$. In order to draw from such RBMs, we make use of Gibbs sampling and burn the first 1000 samples. Additionally, $X \sim \mathcal{N}(Y^1, 0.25 I_2)$ and $A$ follows a Bernoulli distribution with log odds $\frac{1}{5}\sum_{i=1}^{5}(X_i - 0.5)$.

We estimate $\pi$ with the default *LogisticRegression* from the *scikit-learn* package with $C = 1e5$ and *max_iter* = 1000, and $\beta$ with Conditional Mean Embeddings (CME) with the radial basis function kernel and regularization parameter $1e-3$. Figure 3 exhibits the values of $g_n$ over a grid $\{(\theta_1, \theta_2) : \theta_1, \theta_2 \in \{-5, -4.9, -4.8 \ldots, 5\}\}$. Figure 6 exhibits the values of $g_n$ for further pairs $(\theta_1, \theta_2)$, illustrating the behaviour of $g_n$ with different magnitudes of $\theta_1$ and $\theta_2$.

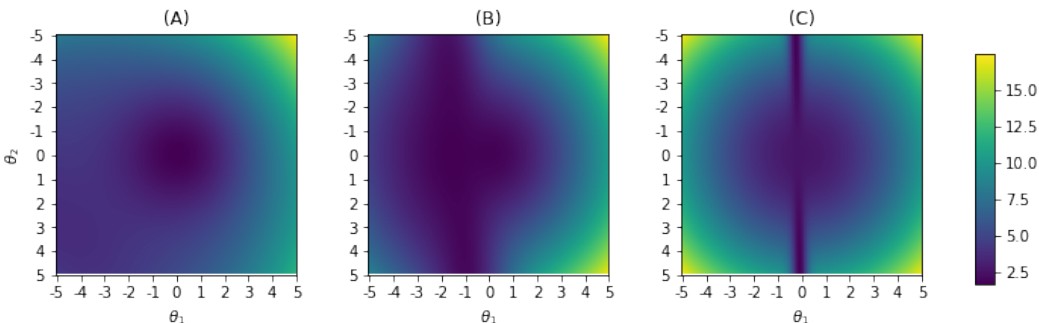

Figure 6: Values of the statistic $g_n(\theta)$ for the RBM with (A) $\theta_1 = -0.1, \theta_2 = 5$, (B) $\theta_1 = -1, \theta_2 = 0.1$, (C) $\theta_1 = 1, \theta_2 = 0.1$.

## B.4 EXPERIMENTAL DATA

We start by training a dense neural network with layers of size [784, 100, 20, 10] on the MNIST train dataset. For this, we minimize the log entropy loss on 80% of such train data and we store the parameters that minimize the log entropy loss for the remaining validation data (remaining 20% of the MNIST train dataset).

We then evaluate this trained neural network on the MNIST test dataset. For each digit, we take the 20-dimensional layer to be $Y^1$. The covariates $X$ are defined as $Y^1 + G$, where $G \sim \mathcal{N}(0, I_{20})$. Treatment $A$ follows a Bernoulli distribution with probability 0.5.

We consider the intractable model $q_\theta(Y^1) \propto \exp(\langle T(Y^1), \theta \rangle)$, where the lower dimensional representation $T(Y^1)$ is taken as the evaluation of the 10-dimensional layer of the neural network. Note that $T$ is therefore differentiable, so we can apply the DR-MKSD estimator.

Out of the whole MNIST test dataset, we take the first 500 observations to train the DR-MKSD estimator (i.e., find $\theta_n$) with $(X_i, A_i, Y_i)_{i=1}^{500}$. We define $\pi = 0.5$ and estimate $\beta$ with Conditional Mean Embeddings (CME) with the radial basis function kernel and regularization parameter $1e - 3$.

In order to obtain Figure 4, we take the minimizers of the estimated unnormalized density of $Y^1$ (the evaluation of the 20-dimensional layer) for the remaining observations of the MNIST test dataset. The top row images are taken randomly for the same subset of the MNIST test dataset.

## C  PROOFS

We now present the proofs of the theorems stated in the main body of the paper. For this, we start by introducing the notation that will be used throughout. The proofs of Theorem 1 and Theorem 2 subsequently follow.

### C.1  NOTATION

Given a Hilbert space $\mathcal{K}$ and a $\mathcal{K}$-valued function $\phi(z) \in \mathcal{K}$, we denote its norm in the Hilbert space by $\|\phi(z)\|_{\mathcal{K}}$. Furthermore, $\|\widehat{\phi}\|^2 = \int \|\widehat{\phi}(z)\|_{\mathcal{K}}^2 \, dP(z)$ denotes the squared $L_2(Q)$ norm of the $\mathcal{K}$-valued function norm. We highlight that the expectation is only taken with respect to the randomness of $Z$, while $\hat{\phi}$ is considered to be fixed. Note that in the case $\mathcal{K} = \mathbb{R}$, $\|\cdot\|^2$ denotes the usual $L_2(Q)$ norm of a real-valued function.

We let $P_n \hat{\phi} = P_n\{\hat{\phi}(Z)\} = \frac{1}{n}\sum_i \phi(Z_i)$ denote the sample average and $P(\hat{\phi}) = P\{\hat{\phi}(Z)\} = \int \hat{\phi}(z) \, dP(z)$ the expected value of $\hat{\phi}$ (Bochner integral) with respect to $Z$, treating $\hat{\phi}$ as fixed. If a function $k$ takes two arguments (this is the case when $k$ is a kernel), then we denote $P_n^2 k = \frac{1}{n^2}\sum_{i,j} k(Y_i, Y_j)$. Further, $Tr$ denotes the trace of a matrix.

Lastly, we make use of standard big-oh and little-oh notation, where $X_n = O_P(r_n)$ implies that the ratio $X_n/r_n$ is bounded in probability, and $X_n = o_P(r_n)$ indicates that $X_n/r_n$ converges in probability to 0. Throughout, we make use of 'calculus' with this stochastic order notation, such as $o_P(1) + O_P(1) = O_P(1)$ and $o_P(1)O_P(1) = o_P(1)$.

### C.2  PROOF OF THEOREM 1

We prove the theorem in three steps:

1. First, we show that $\sup_{\theta \in \Theta} \|P_n \hat{\phi}_\theta - P_n \phi_\theta\|_{\mathcal{H}^d} \xrightarrow{p} 0$.

2. We then prove that $\sup_{\theta \in \Theta} |g_n(\theta) - \mathrm{KSD}(Q_\theta \| Q^1)| \xrightarrow{p} 0$.

3. We finally arrive to $\mathrm{KSD}(Q_{\theta_n} \| Q^1) \xrightarrow{p} \min_{\theta \in \Theta} \mathrm{KSD}(Q_\theta \| Q^1)$.

We now need to demonstrate the validity of each of the steps.

**Details of step 1.** For simplicity, let us assume that $n$ is even and hence $n//2 = n/2$. We note that

$$
\begin{aligned}
P_n \hat{\phi}_\theta - P_n \phi_\theta &= \frac{1}{n}\sum_{i=1}^{n}\left\{\hat{\phi}_\theta(Z_i) - \phi_\theta(Z_i)\right\} \\
&= \frac{1}{n}\left(\sum_{i=1}^{n/2}\left\{\hat{\phi}_\theta(Z_i) - \phi_\theta(Z_i)\right\} + \sum_{i=n/2}^{n}\left\{\hat{\phi}_\theta(Z_i) - \phi_\theta(Z_i)\right\}\right) \\
&= \frac{1}{2}\left(\frac{1}{n/2}\sum_{i=1}^{n/2}\left\{\hat{\phi}_\theta(Z_i) - \phi_\theta(Z_i)\right\}\right) + \frac{1}{2}\left(\frac{1}{n/2}\sum_{i=n/2}^{n}\left\{\hat{\phi}_\theta(Z_i) - \phi_\theta(Z_i)\right\}\right). \\
&= \frac{1}{2}\Bigg(\underbrace{\frac{2}{n}\sum_{i=1}^{n/2}\left\{\hat{\phi}_\theta^{(1)}(Z_i) - \phi_\theta(Z_i)\right\}}_{(I)}\Bigg) + \frac{1}{2}\Bigg(\underbrace{\frac{2}{n}\sum_{i=n/2}^{n}\left\{\hat{\phi}_\theta^{(2)}(Z_i) - \phi_\theta(Z_i)\right\}}_{(II)}\Bigg).
\end{aligned}
$$

Let us first work with term (I), and note that it may be rewritten as $P_{n/2}\hat{\phi}_\theta^{(1)} - P_{n/2}\phi_\theta$. Additionally, $P_{n/2}\hat{\phi}_\theta^{(1)} - P_{n/2}\phi_\theta = T_1(\theta) + T_2(\theta)$, where $T_1(\theta) = (P_{n/2} - Q)(\hat{\phi}_\theta^{(1)} - \phi_\theta)$ is the *empirical process term* and $T_2(\theta) = Q(\hat{\phi}_\theta^{(1)} - \phi_\theta)$ is the *bias term*. Given that $\hat{\phi}_\theta^{(1)}$ was trained independently

from $\{Z_{n/2}, \ldots, Z_n\}$, following Martinez-Taboada et al. (2023, Lemma C.7) and the arguments in Martinez-Taboada et al. (2023, Theorem C.9), we deduce that

$$\|T_1(\theta)\|_{\mathcal{H}^d} = O_P\left(\frac{\|\hat{\phi}_\theta^{(1)} - \phi_\theta\|}{\sqrt{n/2}}\right), \quad \|T_2(\theta)\|_{\mathcal{H}^d} \le \frac{1}{\epsilon}\|\pi - \hat{\pi}^{(1)}\|\|\beta_\theta - \hat{\beta}_\theta^{(1)}\|.$$

Given Conditions (i)-(ii), we deduce that

$$
\begin{aligned}
\sup_{\theta \in \Theta} \|T_1(\theta)\|_{\mathcal{H}^d} + \|T_2(\theta)\|_{\mathcal{H}^d} &\le \sup_{\theta \in \Theta} \|T_1(\theta)\|_{\mathcal{H}^d} + \sup_{\theta \in \Theta} \|T_2(\theta)\|_{\mathcal{H}^d} \\
&= O_P\left(\frac{\sup_{\theta \in \Theta} \|\hat{\phi}_\theta^{(1)} - \phi_\theta\|}{\sqrt{n/2}}\right) + \sup_{\theta \in \Theta}\left\{\frac{1}{\epsilon}\|\pi - \hat{\pi}^{(1)}\|\|\beta_\theta - \hat{\beta}_\theta^{(1)}\|\right\} \\
&= o_P(1) + o_P(1) \\
&= o_P(1).
\end{aligned}
$$

Hence, we infer that $\sup_{\theta \in \Theta} \|P_{n/2}\hat{\phi}_\theta^{(1)} - P_{n/2}\phi_\theta\|_{\mathcal{H}^d} = o_P(1)$. Analogously, we deduce that $\sup_{\theta \in \Theta} \|(II)\|_{\mathcal{H}^d} = o_P(1)$. We thus conclude

$$
\begin{aligned}
\sup_{\theta \in \Theta} \|P_n\hat{\phi}_\theta - P_n\phi_\theta\|_{\mathcal{H}^d} &= \sup_{\theta \in \Theta} \|\frac{1}{2}(I) + \frac{1}{2}(II)\|_{\mathcal{H}^d} \\
&\le \sup_{\theta \in \Theta} \|\frac{1}{2}(I)\|_{\mathcal{H}^d} + \sup_{\theta \in \Theta} \|\frac{1}{2}(II)\|_{\mathcal{H}^d} \\
&= o_P(1) + o_P(1) \\
&= o_P(1).
\end{aligned}
$$

**Details of step 2.** Denoting $\chi_n(\theta) = P_n\hat{\phi}_\theta - P_n\phi_\theta$, we have that

$$
\begin{aligned}
g_n(\theta; Y_1, \ldots, Y_n) &= \left\langle \frac{1}{n}\sum_{i=1}^n \hat{\phi}_\theta(Z_i), \frac{1}{n}\sum_{j=1}^n \hat{\phi}_\theta(Z_j) \right\rangle_{\mathcal{H}^d} \\
&= \left\langle P_n\hat{\phi}_\theta, P_n\hat{\phi}_\theta \right\rangle_{\mathcal{H}^d} \\
&= \langle P_n\phi_\theta + \chi_n(\theta), P_n\phi_\theta + \chi_n(\theta)\rangle_{\mathcal{H}^d} \\
&= \langle P_n\phi_\theta, P_n\phi_\theta\rangle_{\mathcal{H}^d} + 2\langle P_n\phi_\theta, \chi_n(\theta)\rangle_{\mathcal{H}^d} + \langle \chi_n(\theta), \chi_n(\theta)\rangle_{\mathcal{H}^d}.
\end{aligned}
$$

We now work with these three terms separately. First, note that assumption A5 implies that, for all $\theta$ in $\Theta$,

$$\int h_\theta^*(z, z)dP(z) < \int \sup_{\tilde{\theta} \in \Theta} h_{\tilde{\theta}}^*(z, z)dP(z) < \infty,$$

Hence, by Jensen's inequality,

$$\int \sqrt{h_\theta^*(z, z)}dP(z) \le \sqrt{\int h_\theta^*(z, z)dP(z)} < \infty \quad \forall \theta \in \Theta.$$

Given that we have also assumed Conditions A6-A7, we can apply Oates et al. (2022, Lemma 11) to establish that

$$\sup_{\theta \in \Theta} |\langle P_n\phi_\theta, P_n\phi_\theta\rangle_{\mathcal{H}^d} - \mathrm{KSD}(Q_{\theta_n}\|Q^1)| = o_P(1).$$

Second, we have that $\|P_n\phi_\theta\|_{\mathcal{H}^d}^2 = P_n^2\{h_\theta^*\} \le P_n^2\{\sup_{\theta \in \Theta} h_\theta^*\}$, hence $\|P_n\phi_\theta\|_{\mathcal{H}^d}^2 \le P_n^2\{\sup_{\theta \in \Theta} h_\theta^*\}$. Based on the law of large numbers for V-statistics and Conditions A4-A5, we deduce that $P_n^2\{\sup_{\theta \in \Theta} h_\theta^*\} \xrightarrow{p} \iint \sup_{\theta \in \Theta} h_\theta^*(z, \tilde{z})dP(z)dP(\tilde{z}) < \infty$, so

$$\sup_{\theta \in \Theta} \|P_n\phi_\theta\|_{\mathcal{H}^d} = O_P(1). \tag{8}$$

Further, we remind that we have proven $\sup_{\theta \in \Theta} \|\chi_n(\theta)\|_{\mathcal{H}^d} \xrightarrow{p} 0$ on Step 1. Consequently,

$$
\begin{aligned}
\sup_{\theta \in \Theta} |\langle P_n \phi_\theta, \chi_n(\theta) \rangle_{\mathcal{H}^d}| &\overset{(i)}{\leq} \sup_{\theta \in \Theta} \left( \|P_n \phi_\theta\|_{\mathcal{H}^d} \|\chi_n(\theta)\|_{\mathcal{H}^d} \right) \\
&\leq \left( \sup_{\theta \in \Theta} \|P_n \phi_\theta\|_{\mathcal{H}^d} \right) \left( \sup_{\theta \in \Theta} \|\chi_n(\theta)\|_{\mathcal{H}^d} \right) \\
&= O_P(1) o_P(1) \\
&= o_P(1),
\end{aligned}
$$

where (i) is obtained by Cauchy-Schwartz inequality. Lastly, we highlight that

$$
\sup_{\theta \in \Theta} \|\chi_n(\theta)\|_{\mathcal{H}^d} = o_P(1) \implies \sup_{\theta \in \Theta} \|\chi_n(\theta)\|_{\mathcal{H}^d}^2 = o_P(1).
$$

It suffices to note that $\sup_{\theta \in \Theta} |g_n(\theta) - \mathrm{KSD}(Q_{\theta_n} \| Q^1)|$ may be rewritten as

$$
\sup_{\theta \in \Theta} |\langle P_n \phi_\theta, P_n \phi_\theta \rangle_{\mathcal{H}^d} + 2 \langle P_n \phi_\theta, \chi_n(\theta) \rangle_{\mathcal{H}^d} + \langle \chi_n(\theta), \chi_n(\theta) \rangle_{\mathcal{H}^d} - \mathrm{KSD}(Q_\theta \| Q^1)|,
$$

which is upper bounded by

$$
\sup_{\theta \in \Theta} |\langle P_n \phi_\theta, P_n \phi_\theta \rangle_{\mathcal{H}^d} - \mathrm{KSD}(Q_\theta \| Q^1)| + 2 \sup_{\theta \in \Theta} |\langle P_n \phi_\theta, \chi_n(\theta) \rangle_{\mathcal{H}^d}| + \sup_{\theta \in \Theta} |\langle \chi_n(\theta), \chi_n(\theta) \rangle_{\mathcal{H}^d},
$$

which is $o_P(1)$, concluding

$$
\sup_{\theta \in \Theta} |g_n(\theta) - \mathrm{KSD}(Q_\theta \| Q^1)| = o_P(1).
$$

**Details of step 3.** In order to conclude, we extend the argument presented in Oates et al. (2022, Lemma 7) to convergence in probability. Take $\theta^* \in \arg\min \mathrm{KSD}(Q_\theta \| Q^1)$. Given that $\sup_{\theta \in \Theta} |g_n(\theta) - \mathrm{KSD}(Q_\theta \| Q^1)| = o_P(1)$, for any $\epsilon > 0$ and $\delta > 0$, there exists $n^* \in \mathbb{N}$ such that $|g_n(\theta) - \mathrm{KSD}(Q_\theta \| Q^1)| < \frac{\epsilon}{2}$ with probability at least $1 - \delta$ for all $n \geq n^*$. Consequently,

$$
\mathrm{KSD}(Q_{\theta^*} \| Q^1) \leq \mathrm{KSD}(Q_{\theta_n} \| Q^1) + \frac{\epsilon}{2} \leq g_n(\theta_n) \leq g_n(\theta^*) + \frac{\epsilon}{2} \leq \mathrm{KSD}(Q_\theta^* \| Q^1) + \epsilon
$$

for $n \geq n^*$ with probability $1 - \delta$. This is, $|g_n(\theta_n) - \mathrm{KSD}(Q_\theta^* \| Q^1)| < \epsilon$ for $n \geq n^*$ with probability $1 - \delta$, concluding that $g_n(\theta_n) \xrightarrow{p} \mathrm{KSD}(Q_{\theta^*} \| Q^1)$.

### C.3 PROOF OF THEOREM 2

We prove the theorem in three steps:

1. First, we prove $\sup_{\theta \in \Theta} \|P_n \hat{\phi}_\theta - P_n \phi_\theta\|_{\mathcal{H}^d} = o_P\left(\frac{1}{\sqrt{n}}\right)$, and
   $\sup_{\theta \in \Theta} \|\frac{\partial}{\partial \theta} \left\{ P_n \hat{\phi}_\theta - P_n \phi_\theta \right\}\|_{\mathcal{H}^d} = o_P\left(\frac{1}{\sqrt{n}}\right)$.

2. Second, we show that $\left\langle \frac{\partial}{\partial \theta} \left\{ P_n \phi_{\theta_n} \right\}, P_n \phi_{\theta_n} \right\rangle_{\mathcal{H}^d} = \Delta_n / 2$, with $\|\Delta_n\|_{\mathbb{R}^p} = o_P\left(\frac{1}{\sqrt{n}}\right)$.

3. We then conclude that $\sqrt{n}(\theta_n - \theta^*) \xrightarrow{d} N(0, 4\Gamma^{-1} \Sigma \Gamma^{-1})$.

We now need to demonstrate the validity of each of the steps.

**Details of step 1.** With an analogous argument used in Proof C.2 (step 1), we yield

$$
P_n \hat{\phi}_\theta - P_n \phi_\theta = \frac{1}{2} \left( \underbrace{\frac{2}{n} \sum_{i=1}^{n/2} \left\{ \hat{\phi}_\theta^{(1)}(Z_i) - \phi_\theta(Z_i) \right\}}_{(I)} \right) + \frac{1}{2} \left( \underbrace{\frac{2}{n} \sum_{i=n/2}^{n} \left\{ \hat{\phi}_\theta^{(2)}(Z_i) - \phi_\theta(Z_i) \right\}}_{(II)} \right),
$$

where

$$\sup_{\theta \in \Theta} \|(I)\|_{\mathcal{H}^d} = O_P \left( \frac{\sup_{\theta \in \Theta} \|\hat{\phi}_\theta^{(1)} - \phi_\theta\|}{\sqrt{n/2}} \right) + \sup_{\theta \in \Theta} \left\{ \frac{1}{\epsilon} \|\pi - \hat{\pi}^{(1)}\| \|\beta_\theta - \hat{\beta}_\theta^{(1)}\| \right\},$$

$$\sup_{\theta \in \Theta} \|(II)\|_{\mathcal{H}^d} = O_P \left( \frac{\sup_{\theta \in \Theta} \|\hat{\phi}_\theta^{(2)} - \phi_\theta\|}{\sqrt{n/2}} \right) + \sup_{\theta \in \Theta} \left\{ \frac{1}{\epsilon} \|\pi - \hat{\pi}^{(2)}\| \|\beta_\theta - \hat{\beta}_\theta^{(2)}\| \right\}.$$

Assumptions (ii') and (iii') imply

$$\sup_{\theta \in \Theta} \|(I)\|_{\mathcal{H}^d} = o_P \left( \frac{1}{\sqrt{n/2}} \right) + o_P \left( \frac{1}{\sqrt{n}} \right), \quad \sup_{\theta \in \Theta} \|(II)\|_{\mathcal{H}^d} = o_P \left( \frac{1}{\sqrt{n/2}} \right) + o_P \left( \frac{1}{\sqrt{n}} \right),$$

so

$$\sup_{\theta \in \Theta} \|P_n \hat{\phi}_\theta - P_n \phi_\theta\|_{\mathcal{H}^d} = o_P \left( \frac{1}{\sqrt{n/2}} \right) + o_P \left( \frac{1}{\sqrt{n}} \right) = o_P \left( \frac{1}{\sqrt{n}} \right).$$

Similarly, based on Assumptions (ii') and (iii'), we obtain $\sup_{\theta \in \Theta} \|\frac{\partial}{\partial \theta} \left\{ P_n \hat{\phi}_\theta - P_n \phi_\theta \right\} \|_{(\mathcal{H}^d)^p} = o_P \left( \frac{1}{\sqrt{n}} \right)$.

**Details of step 2.** Given that $\theta_n$ is the minimizer of a differentiable function $g_n$, we have that $g'(\theta_n) = 0$. Denoting $\chi_n(\theta) = P_n \hat{\phi}_\theta - P_n \phi_\theta$, we have that

$$
\begin{aligned}
0 &= g'(\theta_n) \\
&= \frac{\partial}{\partial \theta} g(\theta_n) \\
&= \frac{\partial}{\partial \theta} \langle P_n \phi_{\theta_n} + \chi_n(\theta_n), P_n \phi_{\theta_n} + \chi_n(\theta_n) \rangle_{\mathcal{H}^d} \\
&= \left\langle \frac{\partial}{\partial \theta} \left\{ P_n \phi_{\theta_n} + \chi_n(\theta_n) \right\}, P_n \phi_{\theta_n} + \chi_n(\theta_n) \right\rangle_{\mathcal{H}^d} + \\
&\quad + \left\langle P_n \phi_{\theta_n} + \chi_n(\theta_n), \frac{\partial}{\partial \theta} \left\{ P_n \phi_{\theta_n} + \chi_n(\theta_n) \right\} \right\rangle_{\mathcal{H}^d} \\
&= 2 \left\langle \frac{\partial}{\partial \theta} \left\{ P_n \phi_{\theta_n} \right\}, P_n \phi_{\theta_n} \right\rangle_{\mathcal{H}^d} + 2 \left\langle \frac{\partial}{\partial \theta} \left\{ \chi_n(\theta_n) \right\}, \chi_n(\theta_n) \right\rangle_{\mathcal{H}^d} + \\
&\quad + \left\langle P_n \phi_{\theta_n}, \frac{\partial}{\partial \theta} \left\{ \chi_n(\theta_n) \right\} \right\rangle_{\mathcal{H}^d} + \left\langle \frac{\partial}{\partial \theta} \left\{ P_n \phi_{\theta_n} \right\}, \chi_n(\theta_n) \right\rangle_{\mathcal{H}^d}.
\end{aligned} \tag{9}
$$

Let us upper bound the last three terms of the latter addition. First, we note that

$$
\begin{aligned}
\left\| \left\langle \frac{\partial}{\partial \theta} \left\{ \chi_n(\theta_n) \right\}, \chi_n(\theta_n) \right\rangle_{\mathcal{H}^d} \right\|_{\mathbb{R}^p} &\leq \sup_{\theta \in \Theta} \left\| \left\langle \frac{\partial}{\partial \theta} \left\{ \chi_n(\theta) \right\}, \chi_n(\theta) \right\rangle_{\mathcal{H}^d} \right\|_{\mathbb{R}^p} \\
&\overset{(i)}{\leq} \sup_{\theta \in \Theta} \left\{ \| \frac{\partial}{\partial \theta} \left\{ \chi_n(\theta) \right\} \|_{(\mathcal{H}^d)^p} \| \chi_n(\theta) \|_{\mathcal{H}^d} \right\} \\
&\leq \sup_{\theta \in \Theta} \left\{ \| \frac{\partial}{\partial \theta} \left\{ \chi_n(\theta) \right\} \|_{(\mathcal{H}^d)^p} \right\} \sup_{\theta \in \Theta} \left\{ \| \chi_n(\theta) \|_{\mathcal{H}^d} \right\} \\
&= o_P \left( \frac{1}{\sqrt{n}} \right) o_P \left( \frac{1}{\sqrt{n}} \right) \\
&= o_P \left( \frac{1}{n} \right),
\end{aligned} \tag{10}
$$

where (i) is obtained by Cauchy-Schwarz inequality. Second,

$$
\begin{aligned}
\left\| \left\langle P_n\phi_{\theta_n}, \frac{\partial}{\partial\theta}\{\chi_n(\theta_n)\} \right\rangle_{\mathcal{H}^d} \right\|_{\mathbb{R}^p} &\leq \sup_{\theta\in\Theta} \left\| \left\langle P_n\phi_\theta, \frac{\partial}{\partial\theta}\{\chi_n(\theta)\} \right\rangle_{\mathcal{H}^d} \right\|_{\mathbb{R}^p} \\
&\overset{(i)}{\leq} \sup_{\theta\in\Theta} \left\{ \|P_n\phi_\theta\|_{\mathcal{H}^d} \|\frac{\partial}{\partial\theta}\{\chi_n(\theta)\}\|_{(\mathcal{H}^d)^p} \right\} \\
&\leq \sup_{\theta\in\Theta} \{\|P_n\phi_\theta\|_{\mathcal{H}^d}\} \sup_{\theta\in\Theta} \left\{ \|\frac{\partial}{\partial\theta}\{\chi_n(\theta)\}\|_{(\mathcal{H}^d)^p} \right\} \\
&\overset{(ii)}{=} O_P(1) o_P\left(\frac{1}{\sqrt{n}}\right) \\
&= o_P\left(\frac{1}{\sqrt{n}}\right),
\end{aligned}
\tag{11}
$$

where (i) is obtained based on Cauchy-Schwartz inequality, and (ii) is derived as in Equation equation 8, given Conditions A4-A5.

Third, we have that

$$
\begin{aligned}
\|\frac{\partial}{\partial\theta}\{P_n\phi_{\theta_n}\}\|^2_{(\mathcal{H}^d)^p} &\leq \sup_{\theta\in\Theta} \|\frac{\partial}{\partial\theta}\{P_n\phi_\theta\}\|^2_{(\mathcal{H}^d)^p} \\
&= \sup_{\theta\in\Theta} \|P_n\left\{\frac{\partial}{\partial\theta}\phi_\theta\right\}\|^2_{(\mathcal{H}^d)^p} \\
&= \sup_{\theta\in\Theta} \sum_{k=1}^p \|P_n\left\{\frac{\partial}{\partial\theta_k}\phi_\theta\right\}\|^2_{\mathcal{H}^d} \\
&= \sup_{\theta\in\Theta} \sum_{k=1}^p P_n^2\left\{\frac{\partial^2}{\partial\theta_k^2}h_\theta^*\right\} \\
&= \sup_{\theta\in\Theta} P_n^2\left\{\sum_{k=1}^p \frac{\partial^2}{\partial\theta_k^2}h_\theta^*\right\} \\
&\leq P_n^2\left\{\sup_{\theta\in\Theta} \sum_{k=1}^p \frac{\partial^2}{\partial\theta_k^2}h_\theta^*\right\} \\
&= P_n^2\left\{\sup_{\theta\in\Theta} Tr\left(\frac{\partial^2}{\partial\theta^2}h_\theta^*\right)\right\}
\end{aligned}
$$

Note that $Tr\left(\frac{\partial^2}{\partial\theta^2}h_\theta^*\right)$ is dominated by the $L^1$ norm (sum of the absolute values of the entries) of the Hessian $\frac{\partial^2}{\partial\theta^2}h_\theta^*$. Further, the $L^1$ norm is equivalent to the $L^2$ norm $\|\cdot\|_{\mathbb{R}^{p\times p}}$ (all norms are equivalent in finite dimensional Banach spaces). Hence, based on the law of large numbers for V-statistics and Conditions A12-A13, we deduce that

$$
P_n^2\left\{\sup_{\theta\in\Theta} \|\frac{\partial^2}{\partial\theta^2}h_\theta^*\|_{\mathbb{R}^{p\times p}}\right\} \overset{p}{\to} \iint \sup_{\theta\in\Theta} \|\frac{\partial^2}{\partial\theta^2}h_\theta^*(z,\tilde{z})\|_{\mathbb{R}^{p\times p}} dP(z)dP(\tilde{z}) < \infty.
$$

Consequently,

$$
\sup_{\theta\in\Theta} \|\frac{\partial}{\partial\theta}\{P_n\phi_{\theta_n}\}\|^2_{(\mathcal{H}^d)^p} = O_P(1),
$$

and hence

$$
\begin{aligned}
\left\| \left\langle \frac{\partial}{\partial \theta} \{P_n \phi_{\theta_n}\}, \chi_n(\theta_n) \right\rangle_{\mathcal{H}^d} \right\|_{\mathbb{R}^p}
&\leq \sup_{\theta \in \Theta} \left\| \left\langle \frac{\partial}{\partial \theta} \{P_n \phi_\theta\}, \chi_n(\theta) \right\rangle_{\mathcal{H}^d} \right\|_{\mathbb{R}^p} \\
&\overset{(i)}{\leq} \sup_{\theta \in \Theta} \left\{ \left\| \frac{\partial}{\partial \theta} \{P_n \phi_\theta\} \right\|_{(\mathcal{H}^d)^p} \|\chi_n(\theta)\|_{\mathcal{H}^d} \right\} \\
&\leq \sup_{\theta \in \Theta} \left\{ \left\| \frac{\partial}{\partial \theta} \{P_n \phi_\theta\} \right\|_{(\mathcal{H}^d)^p} \right\} \sup_{\theta \in \Theta} \left\{ \|\chi_n(\theta)\|_{\mathcal{H}^d} \right\} \\
&= O_P(1) o_P \left( \frac{1}{\sqrt{n}} \right) \\
&= o_P \left( \frac{1}{\sqrt{n}} \right),
\end{aligned}
\tag{12}
$$

where (i) is obtained based on Cauchy-Schwartz inequality.

Combining Equation equation 9 with the upper bounds from Equations equation 10, equation 11, and equation 12, we derive that

$$
2 \left\langle \frac{\partial}{\partial \theta} \{P_n \phi_{\theta_n}\}, P_n \phi_{\theta_n} \right\rangle_{\mathcal{H}^d} = \Delta_n,
$$

with $\|\Delta_n\|_{\mathbb{R}^p} = o_P \left( \frac{1}{\sqrt{n}} \right) + o_P \left( \frac{1}{\sqrt{n}} \right) + o_P \left( \frac{1}{n} \right) = o_P \left( \frac{1}{\sqrt{n}} \right)$.

**Details of step 3.**

Further, denoting $g_n^*(\theta) := \langle P_n \phi_\theta, P_n \phi_\theta \rangle_{\mathcal{H}^d} \in \mathbb{R}$, we arrive to

$$
2 \left\langle \frac{\partial}{\partial \theta} \{P_n \phi_{\theta_n}\}, P_n \phi_{\theta_n} \right\rangle_{\mathcal{H}^d} = \frac{\partial}{\partial \theta} \langle P_n \phi_{\theta_n}, P_n \phi_{\theta_n} \rangle_{\mathcal{H}^d} = \frac{\partial}{\partial \theta} g_n^*(\theta_n).
$$

Based on the mean value theorem for convex open sets, there exists $\tilde{\theta}_n = t_n \theta_n + (1 - t_n)\theta_*$ with $t_n \in [0, 1]$ for which

$$
\frac{\partial}{\partial \theta} g_n^*(\theta_*) + \frac{\partial^2}{\partial \theta^2} g_n^*(\tilde{\theta}_n) \times (\theta_n - \theta_*) = g_n^*(\theta_n) = \Delta_n.
$$

If $\frac{\partial^2}{\partial \theta^2} g_n^*(\tilde{\theta}_n)$ is non-singular, then rewriting this expression we obtain

$$
\sqrt{n}(\theta_n - \theta_*) = \left[ \frac{\partial^2}{\partial \theta^2} g_n^*(\tilde{\theta}_n) \right]^{-1} \left[ \sqrt{n} \Delta_n \right] - \left[ \frac{\partial^2}{\partial \theta^2} g_n^*(\tilde{\theta}_n) \right]^{-1} \left[ \sqrt{n} \frac{\partial}{\partial \theta} g_n^*(\theta_*) \right].
$$

Under slightly weaker assumptions than stated in Theorem 2, Oates et al. (2022, Theorem 12) showed the non-singularity of $\frac{\partial^2}{\partial \theta^2} g_n^*(\tilde{\theta}_n)$ by convergence to the matrix $\Gamma$, as well as

$$
- \left[ \frac{\partial^2}{\partial \theta^2} g_n^*(\tilde{\theta}_n) \right]^{-1} \left[ \sqrt{n} \frac{\partial}{\partial \theta} g_n^*(\theta_*) \right] \overset{d}{\to} \mathcal{N}(0, 4\Gamma^{-1} \Sigma \Gamma^{-1}).
$$

Given that $\|\Delta_n\|_{\mathbb{R}^p} = o_P \left( \frac{1}{\sqrt{n}} \right)$ and $\frac{\partial^2}{\partial \theta^2} g_n^*(\tilde{\theta}_n) \overset{p}{\to} \Gamma$ non-singular, we deduce that

$$
\left\| \left[ \frac{\partial^2}{\partial \theta^2} g_n^*(\tilde{\theta}_n) \right]^{-1} \left[ \sqrt{n} \Delta_n \right] \right\|_{\mathbb{R}^p} = o_P(1),
$$

hence concluding

$$
\sqrt{n}(\theta_n - \theta_*) = \mathcal{N}(0, 4\Gamma^{-1} \Sigma \Gamma^{-1}).
$$

