# OpenReview forum: "Counterfactual Density Estimation using Kernel Stein Discrepancies"
_ICLR.cc/2024/Conference — ICLR 2024 poster_

### Official Review · Reviewer_nhik · 2023-10-27

**Soundness:** 2 fair
**Presentation:** 2 fair
**Contribution:** 3 good
**Rating:** 6
**Confidence:** 2

**Summary:**

This paper proposes a method for estimating counterfactual densities by minimizing a Kernel Stein Discrepancy. It describes an estimator and algorithm, and conducts statistical analysis proving the consistency and asymptotic normality under certain conditions. Experiments on synthetic data are done illustrating the theoretical results and an experiment on MNIST shows that the results are intuitively reasonable and realistic.

**Strengths:**

# Originality and significance #
As far as I know, this is this first work applying Stein kernel discrepancies to counterfactual estimation. It has moderate novelty, transferring previous work on Kernel Stein discrepancies to the causal estimation framework. Causal inference has become a popular research topic, with applications in healthcare, reinforcement learning, and others.

# Quality #
The proposed estimator is intuitive, simple, and has good theoretical properties without overly strong assumptions. Experiments on synthetic data confirm the theory.

# Clarity #
The estimator, algorithm, and theoretical results are presented clearly and thoroughly.

**Weaknesses:**

# Quality #
The number of baselines that the algorithm is compared to is fairly limited, they only include kernel-based estimators. It would strengthen the work to include more categories of counterfactuals, and discuss the pros and cons of all, such as the computation burden.

# Clarity #
The experimental set-up is a bit unclear (see my question below).

# Significance #
The experimental baselines are fairly limited, making it difficult to gauge the performance of the algorithm compared to SOTA.

**Questions:**

1. If the minimizer is unique, then is the estimate consistent?
2. Can the distribution of $Y^0$ affect the experimental results? It is not mentioned in Section 5.
3. In figure 3, what happens if $\theta_1$ and $\theta_2$ are of different signs and magnitudes?

---

> ### Author Response · Authors · 2023-11-15
> **Rebuttal**
>
> We thank the reviewer for the comments. We would like to address the following weaknesses and questions raised by the reviewer.
>
> - The number of baselines that the algorithm is compared to is fairly limited, they only include kernel-based estimators. It would strengthen the work to include more categories of counterfactuals, and discuss the pros and cons of all, such as the computation burden. The experimental baselines are fairly limited, making it difficult to gauge the performance of the algorithm compared to SOTA.
>
> To our knowledge, the proposed estimator is the first one that can handle unnormalized densities in the counterfactual setting (which is precisely what motivated the authors for this contribution). Hence, we found no natural benchmark to compare the empirical performance of the proposed estimator with. We have now emphasized this fact in the first paragraph of the experiments section, including the comment: "Given that this is the first estimator to handle unnormalized densities in the counterfactual setting, we found no natural benchmark to compare the empirical performance of the proposed estimator with. Hence, we focus on illustrating the theoretical properties and exploring the empirical performance of the estimator in a variety of settings."
>
> Further, we have now added the following comment regarding the computational complexity of our approach: "We note that evaluating $g_n$ has a time complexity of $O(n^2)$. Nonetheless, in stark contrast to the methods presented in Kim et al. (2018); Kennedy et al. (2021), the proposed DR-MKSD procedure only requires the nuisance estimators to be fitted once. This enables DR-MKSD to leverage computationally expensive estimators, such as deep neural networks, providing a significant advantage with respect to these previous contributions."
>
> - If the minimizer is unique, then is the estimate consistent?
>
> Not necessarily. As stated in the first paragraph of page 5, "In general, $g_n$ need not be convex with respect to $\theta$, and thus, estimating $\theta_n$ may involve typical non-convex optimization challenges." If the problem is non-convex, the uniqueness of a global minimum does not translate to consistency (this is a very common, if not inevitable, phenomenon).
>
> - Can the distribution of Y^0 affect the experimental results? It is not mentioned in Section 5.
>
> It cannot. We have now added the following comment to the paragraph that follows equation 4: "Note that the Y^0 does not affect neither $\pi$ nor $\beta_\theta$. In fact, the problem could have been presented as a missing outcome data problem, where the data from $Y^0$ is missing. The authors have posed the problem in counterfactual outcome terms simply for motivational purposes."
>
> - In figure 3, what happens if $\theta_1$ and $\theta_2$ are of different signs and magnitudes?
>
> The "width" of the regions where g_n takes low values varies depending on the magnitude of \theta_1 and \theta_2. We have now included an additional plot in Appendix B with further pairs $(\theta_1, \theta_2)$ of different magnitudes. Although this particular aspect of this restricted Boltzmann machine (RBM) is intriguing, we believe that delving further into the details of this specific RBM might divert attention from the primary focus of the example (which is an illustrating application of the proposed KSD estimator).

---

> > ### Comment · Reviewer_nhik · 2023-11-22
> > **Thanks to the authors for your response**
> >
> > After reading the other reviews and rebuttals, I have decided to increase my score from 5 to 6. I think that the authors have addressed my concerns satisfactorily.

---

### Official Review · Reviewer_kcoz · 2023-10-30

**Soundness:** 4 excellent
**Presentation:** 3 good
**Contribution:** 3 good
**Rating:** 8
**Confidence:** 4

**Summary:**

This paper presents a method for estimating the counterfactual density using Kernel Stein's method. The estimated method exhibits doubly robustness w.r.t. nuisance paraeter estimation.

**Strengths:**

1. This paper is technically strong. It presents a complicated theory on semiparametric inference in a comprehensive manner, which is well-written.
2. Related works provide a comprehensive summary of the relevant literature.

**Weaknesses:**

1. This paper could be improved by adding a discussion on the comparison between the KSD-based method and the projecting-based method (e.g., Kennedy et al., 2021). Readers may be interested in understanding the reasons or practical guidelines for choosing the KSD-based method over other methods.
2. I am concerned about the sample complexity of the KSD-based method, as it appears to have a time complexity of $O(n^2)$. Could you please discuss how the time complexity of this method compares to other competitive methods (e.g., Kennedy et al., 2021 or Kim et al., 2018)?
3. In the experiment, it would be interesting to compare it with other competitive methods.
4. It would be great if there were some fixed working examples by specifying $Q_{\theta}$ for better comprehensibility.

**Questions:**

1. Is $w_i(x)$ is known? If so, is it dependent on $A$? I am asking this because I think the quantity in Equation 5 should be dependent on $A$ too.
2. Need more explanation on Equation 6 from Equation 5. Specifically, in Equation 6, what’s the meaning of $[\hat{\beta}_{\theta}(\hat{\beta})]$?

---

> ### Author Response · Authors · 2023-11-15
> **Rebuttal**
>
> We thank the reviewer for the comments. We would like to address the following weaknesses and questions raised by the reviewer.
>
> - This paper could be improved by adding a discussion on the comparison between the KSD-based method and the projecting-based method (e.g., Kennedy et al., 2021).
>
> We fully agree. We have now added the following comment at the end of the first paragraph of the related work section: "We highlight that none of these alternatives can in principle handle families of densities which unknown normalizing constants. For instance, using the general f-divergences or L^p norms in the projection stage of the estimation (Kennedy et al, 2021) requires access to the exact evaluation of the densities, which includes the normalizing constants. This is precisely what motivated the authors of this contribution to explore Kernel Stein discrepancies in the counterfactual setting".
>
> -  I am concerned about the sample complexity of the KSD-based method, as it appears to have a time complexity of O(n^2). Could you please discuss how the time complexity of this method compares to other competitive methods (e.g., Kennedy et al., 2021 or Kim et al., 2018)?
>
> Once the nuisance estimators $\hat\pi$ and $\hat\beta$ have been trained, evaluating g_n has indeed a time complexity of O(n^2). However, the proposed KSD-based method only requires estimating the nuisance estimators $\hat\pi$ and $\hat\beta$ once. In contrast, the general method discussed in Kennedy et al. (2021) requires training nuisance estimators for every possible parameter $\theta$. Further, the method proposed in Kim et al. (2018) requires computing a localized kernel estimator and a regression for each point in the support.
>
> We believe this difference to be key in our approach, as it opens the door to using estimators that are expensive to train (such as neural networks) given that they do not have to be retrained over and over again. We have now added the following comment before Theorem 1 pointing this out: "We note that evaluating $g_n$ has a time complexity of $O(n^2)$. Nonetheless, in stark contrast to the methods presented in Kim et al. (2018); Kennedy et al. (2021), the proposed DR-MKSD procedure only requires the nuisance estimators to be fitted once. This enables DR-MKSD to leverage computationally expensive estimators, such as deep neural networks, providing a significant advantage with respect to these previous contributions."
>
> - In the experiment, it would be interesting to compare it with other competitive methods.
>
> To our knowledge, the proposed estimator is the first one that can handle unnormalized densities in the counterfactual setting (which is precisely what initially motivated us to explore the proposed idea). Hence, we found no natural benchmark to compare the empirical performance of the proposed estimator with.
>
> We have now emphasized this fact in the first paragraph of the experiments section, including the comment: "Given that this is the first estimator to handle unnormalized densities in the counterfactual setting, we found no natural benchmark to compare the empirical performance of the proposed estimator with. Hence, we focus on illustrating the theoretical properties and exploring the empirical performance of the estimator in a variety of settings."
>
> -  It would be great if there were some fixed working examples by specifying Q_\theta for better comprehensibility.
>
> The first three experiments set a fixed Q_\theta to work with (a normal distribution with mean \theta, an intractable model with a specific potential function, and a restricted Boltzmann machine). We are happy to change the wording or the structure of the experiments section if the reviewer believes that it could be presented in a more clear manner.
>
> - Is $w_i(x)$ known? If so, is it dependent on A= I am asking this because I think the quantity in Equation 5 should be dependent on A too.
>
> The weights $w_i(x)$ are not known a priori, they are the weights fitted in the training of $\hat\beta$. We have now changed the notation from $w_i(x)$ to $\hat w_i(x)$ to make this clear. The weights would depend on A, but A is fixed throughout the work to A=1, so there is no need for making this dependence explicit.
>
> - Need more explanation on Equation 6 from Equation 5. Specifically, in Equation 6, what’s the meaning of $\hat\beta_\theta(\hat\beta)$?
>
> We construct $\hat \beta_\theta$ from $\hat \beta$. That is, we do not retrain $\hat\beta_\theta$ for each $\theta$. Instead, we train $\hat\beta$ and build on it to derive $\hat\beta_\theta$. We have now added the following sentence right before equation (6): "In order to avoid refitting the estimator $\hat\beta_\theta$ for each value of $\theta$, we propose to define $\hat\beta_\theta$ from $\hat\beta$ as follows: "

---

> > ### Comment · Reviewer_kcoz · 2023-12-01
> > **Response**
> >
> > I am satisfied with the authors' responses. I don't have any remaining questions. I raised the score from 6->8.

---

### Official Review · Reviewer_bMPy · 2023-11-01

**Soundness:** 3 good
**Presentation:** 3 good
**Contribution:** 3 good
**Rating:** 5
**Confidence:** 3

**Summary:**

Inference causal effect is an important question among many domains. The paper introduces an estimator for modeling counterfactual distributions given a flexible set of distributions that only need to be known up to their normalizing constants. The procedure is based on minimizing the kernel Stein discrepancy between this flexible set and the counterfactual distribution. Simultaneously, it accounts for sampling bias in a doubly robust manner. Besides,  the paper provides theoretical foundation for the consistency and asymptotic normality of the estimator, ensuring its reliability and statistical soundness.

**Strengths:**

-   MKSD estimators have primarily been employed for conducting goodness-of-fit tests and sample quality analysis. Nevertheless, in the counterfactual context of this study, the MKSD estimator had not been previously proposed.
-   Conversely, under certain assumptions, the distribution of either counterfactual can be expressed in terms of observational data. This opens the possibility of using MKSD as the primary tool to address the counterfactual distribution estimation problem.

- The paper is well-organized and maintains a clear, logical flow throughout. The structure makes it easy for readers to follow the research from start to finish.

- The paper  incorporates an abundance of related work. This comprehensive overview of prior research provides valuable context for the study and underscores the authors' deep understanding of the field.

- The abstract and introduction effectively set the stage for the paper's major content. They provide a consistent and coherent overview of the research objectives, making it clear to the reader what to expect in the subsequent sections.

**Weaknesses:**

Numerous other studies have explored semiparametric estimators within the debiased machine learning framework for counterfactual density estimation, such as:

[1] Mou, Wenlong, Martin J. Wainwright, and Peter L. Bartlett. "Off-policy estimation of linear functionals: Non-asymptotic theory for semi-parametric efficiency." _arXiv preprint arXiv:2209.13075_ (2022).

[2] Mou, Wenlong, et al. "Kernel-based off-policy estimation without overlap: Instance optimality beyond semiparametric efficiency." _arXiv preprint arXiv:2301.06240_ (2023).

[3] Kennedy, Edward H., Sivaraman Balakrishnan, and Larry Wasserman. "Semiparametric counterfactual density estimation." _arXiv preprint arXiv:2102.12034_ (2021).

- How do you compare the MKSD estimator with other estimators?
- What are the advantages of using MKSD estimators in comparison to other methods?

**Questions:**

See above "weaknesses" section.

---

> ### Author Response · Authors · 2023-11-15
> **Rebuttal**
>
> We thank the reviewer for the comments. We would like to address the following weaknesses raised by the reviewer.
>
> - How do you compare the MKSD estimator with other estimators? What are the advantages of using MKSD estimators in comparison to other methods?
>
> The ultimate goal of our contribution is to model the counterfactual distribution by a family of densities that need not be specified up to normalizing constants. To our knowledge, none of the other estimators can handle these families of distributions. Hence, the proposed MKSD-based estimator opens the door to considering more flexible families of distributions to model counterfactuals (this is, precisely, what motivated us to explore Kernel Stein discrepancies in the counterfactual setting).
>
> While we highlighted (in the paper) that the proposed estimator can handle these families of distributions, we now recognize that we first did not emphasize that none of the other estimators can be used if the normalizing constants are not known. Hence, we have now added the following comment at the end of the first paragraph of the related work section: "We highlight that none of these alternatives can in principle handle families of densities with unknown normalizing constants. For instance, using the general f-divergences or L^p norms in the projection stage of the estimation (Kennedy et al, 2021) requires access to the exact evaluation of the densities, which includes the normalizing constants. This is precisely what motivated the authors of this contribution to explore Kernel Stein discrepancies in the counterfactual setting".

---

### Official Review · Reviewer_d5Rs · 2023-11-01

**Soundness:** 3 good
**Presentation:** 3 good
**Contribution:** 2 fair
**Rating:** 6
**Confidence:** 4

**Summary:**

Under the assumption of conditional ignorability of a dichotomous treatment, this paper studies counterfactual densities. The densities are estimated within parametric classes and are only estimated up to a constant. The procedure involves doubly robust estimation, cross fitting, and kernel Stein discrepancies. The authors prove asymptotic normality for the finite density parameter with product rate conditions on nuisances.

**Strengths:**

Originality: It appears that the estimator is a new combination of known tools, namely doubly robust estimation, cross fitting, and kernel Stein discrepancies, for a different task than previous work that combined these tools (Lam and Zhang 2023). The closest pieces of work are Fawkes et al. (2022) and Martinez-Taboada et al. (2023), which use kernel mean embeddings instead of kernel Stein discrepancies. All of these papers are well cited.

Quality: The results are generally high quality, though some aspects seemed incomplete; see my comments below.

Clarity: The paper is written very well.

Significance: I would like the authors to provide further discussion in this regard; see my comments and questions below.

**Weaknesses:**

1. In my evaluation, the analysis is a direct extension of Martinez-Taboada et al. (2023), replacing features of Y with the xi object evaluated at Y.

2. I would like to see more discussion of how good the optimization of the parameter must be for these results to be applicable.

3. The paragraph beginning with “we underscore that…” was confusing.

**Questions:**

1. Many doubly robust methods exist for counterfactual distributions. When is it a prohibitively difficult problem to transform a counterfactual distribution to a counterfactual density? This seems central to the paper’s motivation.

2. What are use cases in which we care to recover the parameter of the counterfactual density (up to a constant in some sense) rather than the counterfactual density? This seems central to the paper’s motivation.

3. How do the results provided for the parameter translate to results for the counterfactual density?

3. How is the asymptotic variance estimated?

4. Do the corresponding confidence intervals have the desired coverage in Monte Carlo experiments?

5. Is not Hd the sum kernel rather than the product kernel? I believe the sum kernel is not characteristic, and wonder if that is an issue.

I will improve my score if these items are addressed.

---

> ### Author Response · Authors · 2023-11-15
> **Rebuttal**
>
> We thank the reviewer for the comments. We would like to address the following weaknesses and questions raised.
>
> - In my evaluation, the analysis is a direct extension of Martinez-Taboada et al. (2023).
>
> While both works exploit kernel mean embeddings in a doubly robust manner, we emphasize the profound difference in the technical challenges tackled by these works (the two proofs have very little in common). In order to underscore this difference, we have added the sentence "The main theoretical contribution of Martinez-Taboada et al. (2023) is the extension of cross U-statistics to the causal setting; in stark contrast, our contribution deals with the theoretical properties of the minimizer of a 'debiased' V-statistic." to the paragraph that comes right after equation 7.
>
> -  How good the optimization of the parameter must be for these results to be applicable?
>
> In terms of consistency, it is nearly impossible to state any theoretical property if the optimization problem does not output the global minimum. In terms of asymptotic normality, we note that our proof holds even if $\theta_n$ converges to any stationary point (instead of the actual minimum). We can make this remark if the reviewer believes it is of any interest.
>
> However, a potential lack of consistency does not necessarily pose a problem. We have added the following paragraph to Section 4:
> "Furthermore, a potential lack of consistency does not necessarily translate to a poor performance of the proposed estimator. Two distributions characterized by very different parameters $\theta$ may be close in the space of distributions defined by the KSD metric (analogously to two neural networks with very different weights having similar empirical performance). Estimating the distribution characterized via the parameter, not the parameter itself, is the ultimate goal of this work. Hence, the proposed estimator can yield good models for the counterfactual distribution even with inconsistent $\theta_n$."
>
> - The paragraph beginning with “we underscore that…” was confusing.
>
> We have rewritten the paragraph in an effort to be more precise. We refer the reviewer to the revised version of the paper for the new version of the paragraph. We are happy to go over it again if the reviewer believes that it could still be improved.
>
> - When is it a prohibitively difficult problem to transform a counterfactual distribution to a counterfactual density?
>
> We are unaware of any work that transforms a counterfactual cdf in a counterfactual pdf. The cdf estimators are usually piecewise constant. These are not differentiable, and hence the pdf may not be retrieved by differentiating the estimated cdf. We are happy to add a note in the paper about this if the reviewer finds it appropriate.
>
> - What are use cases in which we care to recover the parameter of the counterfactual density (up to a constant in some sense) rather than the counterfactual density?
>
> We assume that the density is parametrized by \theta, so estimating \theta is equivalent to estimating the density.
>
> - How do the results provided for the parameter translate to results for the counterfactual density?
>
> If the map between the parameter space to the distributional space is smooth, then consistency in the parameter space implies consistency in the distributional one.
>
> - How is the asymptotic variance estimated?
>
> The asymptotic variance can be estimated with a consistent sandwich estimator, as proposed in Oates (2021). We have now added the following comment after Theorem 2 "As proposed in Oates et al. (2022), a sandwich estimator $4 \Gamma_n^{-1} \Sigma_n \Gamma_n^{-1}$ for the asymptotic variance $4 \Gamma^{-1} \Sigma \Gamma{-1}$ can be established (Freedman, 2006), where $\Sigma_n$ and $\Gamma_n$ are obtained by substituting $\theta_*$ by $\theta_n$ and replacing the expectations by empirical averages. If the estimator $4 \Gamma_n^{-1} \Sigma_n \Gamma_n^{-1}$ is consistent at any rate, then $\Sigma_n^{-1/2} \Gamma_n(\theta_* - \theta_n) \stackrel{d}{\to} N(0, 1)$ in view of Slutsky's theorem."
>
> - Do the corresponding confidence intervals have the desired coverage in Monte Carlo experiments?
>
> We have now included an analysis of the empirical coverage of the confidence intervals (CI) in Appendix B. They studied 0.95-CIs present empirical coverages of 0.94 and 0.96.
>
> - I believe the sum kernel is not characteristic, and wonder if that is an issue.
>
> Indeed, H^d is the sum kernel. We stated that H^d is a product RKHS equipped with the sum kernel. Our work inherits the properties of the Kernel Stein discrepancies. We refer the reviewer to Chwialkowski et al. (2016) Section 2.1 for a discussion on necessary conditions for the KSD to fully characterize the distribution. In particular, Chwialkowski et al. (2016) Theorem 2.2 states that if the kernel is cc-universal, then the KSD is 0 if and only if the distributions are the same. In the paper, we refer the reader to Chwialkowski et al. (2016) right after equation (1).

---

### Meta-Review · Area_Chair_1gDg · 2023-12-11

**Metareview:**

This paper proposes to model counterfactual distributions by minimizing kernel Stein discrepancies. The idea consider how to deal with troubles of manipulating energy-based models by using those kernel Stein discrepancies. They propose estimators for a parametric class of distributions, some theoretical analyses, and empirical results. The process is a typical sound one: propose an estimator, prove some properties, experimentally study the approach. The committee is overall positive about the submission.

**Justification For Why Not Higher Score:**

It remains to be seen how much of impact this may have.

**Justification For Why Not Lower Score:**

The work is well thought, designed and performed. The topic matches the conference well.

---

### Decision · Program_Chairs · 2024-01-16

Accept (poster)